# Study on the Microscopic Mechanism and Performance of TPU/SBR Composite-Modified Asphalt

**DOI:** 10.3390/polym16192766

**Published:** 2024-09-30

**Authors:** Li Wei, Linxianzi Li, Mingmei Liang, Hongliu Rong, Xiaolong Yang

**Affiliations:** 1College of Architecture and Civil Engineering, Nanning University, Nanning 541699, China; weili@unn.edu.cn; 2College of Civil Engineering and Architecture, Guangxi University, Nanning 530004, China; l2010391059@163.com (M.L.); ronghongliu@sohu.com (H.R.); xiaolongyang@gxu.edu.cn (X.Y.)

**Keywords:** asphalt pavement, composite-modified asphalt, TPU, SBR, basic performance, micro-properties

## Abstract

To enhance the service life of traditional asphalt pavement and mitigate issues such as high-temperature rutting and low-temperature cracking, this study investigates the composite modification of matrix asphalt using thermoplastic polyurethane (TPU) and styrene-butadiene rubber (SBR). Initially, the study examines the conventional properties of the composite-modified asphalt from a macro perspective, analyzing the performance variations of asphalt before and after TPU and SBR modification. Subsequently, microscopic analysis is conducted to explore the microstructure, phase structure, and modification mechanisms of the composite-modified asphalt, with a focus on understanding the underlying reasons for performance changes. The influence of TPU and SBR on asphalt performance is evaluated comprehensively. It is found that TPU-modified asphalt demonstrates superior high-temperature performance, storage stability, and elastic recovery. Conversely, SBR-modified asphalt excels in ductility at low temperatures, though its storage stability decreases with increasing dosage. Based on a thorough analysis of the conventional properties of the two types of modified asphalt, the optimal dosages of TPU and SBR are determined to be 15% and 3.5%, respectively. In the composite-modified asphalt, TPU facilitates the even distribution of chemical components, creating a more stable cross-linked network structure. The compatibility of TPU, SBR, and asphalt contributes to the good storage stability of the composite-modified asphalt. While SBR effects physical modification, TPU induces chemical modification of asphalt. Consequently, the composite modification system benefits from both physical and chemical enhancements, resulting in excellent overall performance.

## 1. Introduction

Asphalt pavement is widely used in road engineering because of its advantages such as short construction period, smooth surface, and driving comfort [1,2]. However, with the rapid development of the global economy, people’s living standards gradually improve and they travel more frequently; therefore, the traffic load gradually increases. Under the coupling effect of multiple factors such as temperature change, moisture penetration, and traffic load, asphalt pavement appears with ruts, cracks, potholes, and other diseases, which seriously affect driving comfort and safety, shorten the service life of asphalt pavement, and increase maintenance cost [3,4,5]. In order to effectively solve the above problems of asphalt pavement, reduce the damage of asphalt pavement, and improve its road performance, it is necessary to improve the traditional asphalt materials. The common method is to add polymer modifiers to asphalt materials to improve the high-temperature stability and low-temperature crack resistance of asphalt, so as to extend the service life of asphalt pavement and meet the requirements of traffic development.

At present, the polymer modifiers commonly used at home and abroad mainly include thermoplastic elastomers, rubber, and thermoplastic resins, such as SB, SBR, NR, PE, EVA, etc. These modifiers can improve the stability, temperature sensitivity, and cohesivity of asphalt to varying degrees, thus enabling asphalt to have better performance in road surface engineering [6]. Among them, the SBR modifier has a good low-temperature modification effect, and the addition of a small amount of SBR can significantly improve the low-temperature performance and aging resistance of asphalt, and effectively reduce the cost [7,8,9]. However, relevant studies have shown that SBR has no significant effect on the improvement of high-temperature stability, and there is no obvious chemical reaction between SBR and asphalt, so it is difficult to form a stable thermodynamic system with asphalt. The addition of SBR will also destroy the structure of asphalt, reduce the storage stability of modified asphalt [10,11], and further affect the mechanical properties and road performance of modified asphalt, ultimately reducing the service life of asphalt pavement. Lv et al. [12] used SBR and nano ${\mathrm{CaCO}}_{3}$ as composite modifiers to carry out composite modification of Burton rock asphalt (BRA), and analyzed the thermal stability and aging resistance of different composite-modified asphalt. The test results show that the composite modifier has no obvious effect on the thermal stability and anti-aging properties of BRA. Liu Bing [13] modified No. 90 matrix asphalt with 1%, 2%, 3%, 4%, and 5% SBR and studied its performance changes. It is found that SBR can significantly improve the low-temperature performance of asphalt and its mixture, but its effect on the high-temperature performance of asphalt is not obvious and even has a negative impact. In order to improve the low-temperature performance of rock asphalt (BRA), Chen Kai et al. [14] mixed styrene butadiene rubber (SBR) into BRA-modified asphalt to prepare composite-modified asphalt. The content of SBR latex should be between 4% and 6%, which can not only ensure the excellent high-temperature performance of BRA but also enhance its anti-aging performance, so as to achieve the best asphalt performance. From the current study, it can be concluded that SBR latex has obvious improvement in the low-temperature performance of pure asphalt emulsion slag, but the improvement in the high-temperature performance and temperature sensitivity is relatively small, and still can not meet the higher requirements for the performance of road construction materials.

Polyurethane is a booming organic polymer material, which has the dual advantages of plastic and rubber and is widely used in coatings, sealants, waterproof materials, building restoration, etc. At the same time, polyurethane materials have excellent physical and chemical properties and have been gradually applied to pavement engineering in recent years [15]. Compared with other modifiers, polyurethane can chemically react with asphalt to form a chemical cross-linked network structure system, which makes the modified asphalt have good storage stability and effectively improves the comprehensive performance of the modified asphalt [16]. FangYing et al. [17] determined the best preparation process parameters of polyurethane-modified asphalt by orthogonal test and intuitive analysis, which were shear temperature 150 °C, shear rate 40 r/min, and shear time 40 min. The results showed that compared with matrix asphalt, SB-modified asphalt, and SBR-modified asphalt, polyurethane-modified asphalt has the best high-temperature performance. However, its low-temperature performance is slightly lower than that of SBR-modified asphalt [17]. Liu et al. found that hemp oil-based polyurethane could significantly improve the high-temperature performance of bio-asphalt, but the product of chemical reaction between polyurethane and asphalt would have a negative impact on the low-temperature performance of asphalt [18]. The results showed that the polyurethane bond was formed by the reaction between the polyurethane and polar hydroxyl groups. Polyurethane has a good performance in improving the high-temperature performance of asphalt, but its stability and durability under low-temperature conditions have not been significantly improved. Liu Ying et al. [19] showed that polyurethane-modified asphalt is a reactive modified asphalt, which makes it have excellent storage stability. Polyurethane-modified asphalt mixture has excellent heat resistance and better water stability, but at low temperatures, its deformation resistance is worse than that of SB-modified asphalt mixture. Fan Teng et al. [20] showed that polyurethane and rubber powder could significantly improve the high-temperature performance of asphalt, and effectively improve the high-temperature resistance and water resistance of asphalt mixture, but would reduce the low-temperature performance of asphalt. Current research indicates that polyurethane, as a chemically modified material, contains functional groups capable of reacting with and bonding to asphalt. This interaction results in a stable dispersion within the asphalt matrix and enhances the cross-linking effect among asphalt molecules. Consequently, polyurethane significantly improves the high-temperature performance and storage stability of asphalt and its mixtures, demonstrating promising potential for engineering applications. However, polyurethane may adversely affect the low-temperature performance of asphalt, leading to increased brittleness and a higher propensity for cracking during winter conditions, which compromises the structural integrity of asphalt pavements. Therefore, it is essential to identify a complementary modifier that can be blended with polyurethane to address these low-temperature performance issues effectively while also providing economic benefits for composite modification.

Based on an analysis of the current research both domestically and internationally, styrene-butadiene rubber (SBR) is known to significantly enhance the low-temperature performance and fatigue resistance of asphalt and its mixtures. However, its limitations include a lack of notable improvement in high-temperature performance and inadequate storage stability, which constrain its application in regions with high summer temperatures. Consequently, there has been considerable research into the composite modification of asphalt by combining SBR with other modifiers, such as montmorillonite (MMT). While these combinations can address the deficiencies of SBR-modified asphalt in high-temperature conditions, they have not effectively resolved the issue of storage stability. Given the performance advantages of thermoplastic polyurethane (TPU) and styrene-butadiene rubber (SBR), leveraging the physical and chemical blending effects of their combination could significantly enhance the performance and service life of asphalt. However, there is currently a lack of systematic research on the use of TPU and SBR in composite modification of asphalt. This gap includes the absence of a comprehensive performance evaluation system and a material design theory, which impedes the practical application and further development of such composite-modified asphalt. This study investigates the composite modification of matrix asphalt using thermoplastic polyurethane (TPU) and styrene-butadiene rubber (SBR). Specifically, TPU and SBR are employed as individual modifiers to produce TPU-modified asphalt and SBR-modified asphalt, respectively. The research involves basic performance testing of these asphalts to analyze the impact of each modifier on asphalt properties. The optimal dosages of TPU and SBR are determined through these tests, and TPU/SBR composite-modified asphalt is prepared accordingly. The study evaluates storage stability, elastic recovery, microstructure, and modification mechanisms of the modified asphalt using standard experimental methods. Additionally, the influence of the TPU/SBR combination on the properties of the modified asphalt is explored. The research framework and methodology for this study are illustrated in Figure 1. We hope to develop a composite-modified asphalt with more comprehensive performance, providing ideas for the research and development of high-performance and long-life materials for road surfaces.

## 2. Materials and Test Methods

### 2.1. Materials

#### 2.1.1. Asphalt

To thoroughly analyze the impact of TPU and SBR modifiers on the performance of base asphalt, the primary technical performance indices of Kunlun A-70 base asphalt are presented in Table 1.

#### 2.1.2. Thermoplastic Polyurethane (TPU)

Polyurethane is a high molecular weight polymer with interlocking soft and hard segments [21,22]. According to different processing methods and molecular structures, polyurethane can be divided into two types: thermoplastic polyurethane and thermosetting polyurethane [23,24]. The specific type of polyurethane selected in this article is thermoplastic polyurethane, which is white in color and appears as a powdery solid.

The properties of TPU are shown in Table 2.

TPU is prepared by reaction polymerization of glycol and diisocyanic acid. Its structure is mainly composed of the -NH-COO group [25], and its reaction formula is as shown in Equation (1).
(1)—N=C=O—+—OH—→—NH—COO—

#### 2.1.3. Styrene Butadiene Rubber (SBR)

Based on its morphology at ambient temperature, styrene-butadiene rubber (SBR) can be classified into two forms: solid particles and liquid latex. For this study, the selected SBR was in the form of solid powder, specifically SBR1502, which appeared as a white to yellowish powder at room temperature. This material is produced by Shanghai Yuanxiang Industrial Co., Ltd., Shanghai, China. The fundamental properties of the SBR are detailed in Table 3.

#### 2.1.4. Aggregates and Mineral Powder

In this study, diabase from Wuming, Guangxi, was used as the aggregate. The raw aggregate was sieved using a laboratory sieve to separate it into coarse and fine aggregates of varying particle sizes. These aggregates were then tested according to relevant standards and specifications, with the results detailed in Table 4 and Table 5. The data from these tables indicate that the technical specifications of the aggregates comply with the prescribed standards. Additionally, mineral powder was chosen as the filler, and its technical specifications were evaluated (refer to Table 6). The results demonstrate that all tested parameters of the limestone mineral powder adhere to the relevant regulations.

### 2.2. Preparation of Modified Asphalt

#### 2.2.1. Preparation of TPU Modified Asphalt

Based on previous research experience, the TPU powder dosage selected in this study is 0%, 5%, 10%, 15%, and 20%, respectively [26]. Firstly, the base asphalt to the flowing state was heated, the shearing temperature was set to 140 °C, and the high-speed shear was started slowly. Next, a small amount of TPU powder was added to the base asphalt many times by using the additive method and the TPU powder was moved to the center of the shear rotor with the mixing bar until the TPU powder was completely incorporated into the asphalt. Then, the shearing rate of the high-speed shear was adjusted to 3000 rpm and maintain it for 60 min. The shearing rate of the high-speed shear was adjusted to 800 r/min the shear temperature and shear rate were kept constant and were then sheared for 10 min to remove bubbles.

#### 2.2.2. Preparation of SBR-Modified Asphalt

According to the existing research on SBR-modified asphalt, the powder content is finally determined as 2%, 2.5%, 3%, 3.5% and 4% [6,7,8,9,10]. Five portions of base asphalt were heated to a molten state at 150 °C, the shear temperature was adjusted to 140 °C, and the shear rate to 3000 rpm. A small amount of SBR powder was added to the base asphalt many times. After it was added completely, the shearing continued for 60 min and then the shear rate was adjusted to a high-speed shear of 800 r/min, and the shear temperature and shear rate were unchanged. After 10 min it was cut to remove bubbles.

#### 2.2.3. Preparation of TPU/SBR Composite-Modified Asphalt

First, the base asphalt was heated to the molten state at 150 °C, and the shear temperature was adjusted to about 140 °C. Then, SBR powder was added until it is completely incorporated into the asphalt and sheared at 1000 r/min for 10 min. The shear temperature and shear rate were kept unchanged and a small amount of TPU powder was added several times. The shear rate was adjusted to 4000 r/min, and the shearing was continued for 60 min until the SBR adhesive powder and TPU adhesive powder were fully and evenly dispersed in the asphalt. Then, the shear rate of the high-speed shear machine was adjusted to 800 r/min, the shear temperature and shear rate were kept unchanged, and the bubble was removed after 10 min of shear. The preparation process of composite-modified asphalt is shown in Figure 2.

### 2.3. Composition Design of Asphalt Mixture

#### 2.3.1. Mix Design

In this study, the gradation of AC-13 grade asphalt mixture was chosen as the focus of investigation, and the gradation curve is illustrated in Figure 3.

#### 2.3.2. Optimal Asphalt-to-Aggregate Ratio

The optimum oil–stone ratio directly affects the pavement performance of asphalt mixture during its service on asphalt pavement. In this study, asphalt mixture was prepared by the “wet method” and the best oil–stone ratio of different types of asphalt mixture was determined according to the asphalt mixture Marshall test method. Using matrix asphalt as an example, based on previous studies, 5 different types of oil–stone ratios were preliminarily determined: 4.2%, 4.5%, 4.8%, 5.1%, and 5.4%. For each oil–stone ratio, four specimens of asphalt mixture were prepared for the Marshall test. According to this method, the optimum oil–stone ratios of SBR-modified asphalt mixture, TPU-modified asphalt mixture, and TPU/SBR composite-modified asphalt mixture are 5.0%, 5.2%, and 5.3%, respectively.

### 2.4. Test Methods

#### 2.4.1. Basic Performance Tests

(1)Physical properties test

The three indexes can reflect the high-temperature stability and low-temperature crack resistance of asphalt materials, are highly practical, and are widely used in asphalt pavement engineering practice. In order to explore the physical properties of TPU/SBR composite-modified asphalt, according to the technical requirements of JTGE20-1 for polymer-modified asphalt, needle penetration tests were carried out at 25 °C for different asphalt materials, and softening point tests and ductility tests were carried out at 5 °C. The high-temperature performance of asphalt can be characterized by the degree of needle penetration and softening point. The smaller the degree of needle penetration, the higher the softening point, indicating the stronger the high-temperature performance of asphalt [25,26,27]. Ductility represents the resistance of asphalt to plastic deformation and is an important indicator of asphalt low-temperature performance. The larger ductility indicates the better low-temperature crack resistance of asphalt [28].

(2)Storage stability test

In the process of storage, transportation, and construction of modified asphalt, separation and subsidence of the modifier and asphalt may occur, which will affect the road performance of the modified asphalt mixture and further limit the popularization and application of polymer modifiers [29,30]. Therefore, it is necessary to analyze the storage stability of modified asphalt. According to the segregation test requirements of JTGE20-1 polymer-modified asphalt, the softening points of the upper and lower layers of different modified asphalt were tested in this study. First, the asphalt was injected into the aluminum tube, the mass was about 50 g, the open end of the aluminum tube was folded and clamped, and the aluminum tube and bracket were placed in the oven at 163 °C for 48 h. It was then transferred to the refrigerator and cooled for 4 h so that the asphalt sample remained solid. The storage stability of asphalt can be judged by the difference in softening points of the upper and lower layers. The smaller the difference, the better the storage stability of asphalt and the less segregation [31].

(3)Elastic recovery test

In order to study the elastic recovery and deformability of TPU/SBR composite-modified asphalt, the elastic recovery rate of matrix asphalt and modified asphalt was evaluated according to the requirements of JTGE20-1. Firstly, the flowing bitumen was poured into the test mold, cooled at room temperature for 90 min, scraped off the bitumen higher than the test mold with a hot scraper, made flush with the surface of the test mold, put into a constant temperature water bath at 25 °C for 90 min, stretched at a speed of 5 cm/min to 10 cm, and cut with scissors in the middle. The asphalt sample was tested after a 60 min water bath at the same water temperature, and the average value of the three measured values was calculated and taken as the elastic recovery test result. The greater the elastic recovery rate, the better the asphalt resistance to permanent deformation [32].

#### 2.4.2. Micro-Properties Tests

(1)Atomic force microscopy (AFM) test

An atomic force microscope (AFM) test can analyze the micro-surface morphology, viscosity, and molecular interaction of polymer-modified asphalt [25,26,27,28,29,30,31,32,33,34,35]. In this study, a FastScan atomic force microscope was used (from BRUKER Corporation in the United States). The imaging mode was tap type, and the scanning area was set as 20 μm × 20 μm. Two-dimensional and three-dimensional morphologies of different asphalt materials were obtained by scanning. The dissolution and dispersion of modifiers in asphalt were evaluated by analyzing the area, quantity, and height of micro-surface honeycomb structures of different asphalt materials. The compatibility between the modifier and asphalt was analyzed to determine the stability of the newly formed modification system.

The principle of the atomic force microscopy test is shown in Figure 4.

(2)Fourier transform infrared spectroscopy (FTIR) test

Fourier transform infrared spectroscopy (FTIR) test can analyze the chemical composition and functional groups of polymer-modified asphalt [25,26,27,28,29,30,31,32,33,34]. The instrument used is a VERTEX 70 Fourier Transform Infrared Spectrometer (sourced from Bruker GmbH in Karlsruhe, Germany). During the experiment, a scanning resolution of 4 cm^−1^ and a wavelength range of 4000~400 cm^−1^ were used, with 32 scans for each type of asphalt. The prepared asphalt was placed in the corresponding position of the infrared spectrometer, using a computer to collect data and obtain corresponding infrared spectra. Based on the characteristic peaks of different asphalt samples, the molecular structure and functional groups of the matrix asphalt and modified asphalt can be analyzed.

The principle of the Fourier transform infrared spectroscopy (FTIR) test is shown in Figure 5.

(3)Fluorescence microscope (FM) test

Fluorescence microscopy is based on the human eye; therefore, it can not see a certain wavelength of high-energy ultraviolet light used to irradiate the material and activate its internal fluorophores. This activation produces clearly visible fluorescent points that can be examined microscopically through an imaging system, allowing the microscopic phase structure of the material to be clearly observed. Relevant studies have found that the modifier in modified asphalt will produce a fluorescence phenomenon under the excitation of high-energy ultraviolet light, that the compatibility between the modifier and asphalt can be judged by the distribution of fluorescence points in FM images, and that the microscopic phase structure of modified asphalt can be analyzed [36]. The instrument is selected from an IMAGER.Z2 fluorescence microscope (sourced from Carl Zeiss Optics GmbH in Wetzlar, Germany). At present, the FM test sample preparation method adopts the hot drop cover glass method for sample preparation. The magnification of the sample observation is 100 times.

The principle of the fluorescence microscope test is shown in Figure 6.

## 3. Results and Discussion

### 3.1. Basic Performance

#### 3.1.1. Performance Analysis of Single Mixed Modified Asphalt

(1)physical properties

Base asphalt was used as a control group in this study. The physical properties of modified bitumen with TPU contents of 5%, 10%, 15%, and 20% were analyzed. The test results are shown in Figure 7.

When TPU was added to the base asphalt, the penetration of asphalt decreased. When TPU was added at the beginning (the content was 5%), the penetration of asphalt changed the most and decreased by 12.4 mm. With the increase in TPU content, the penetration of asphalt showed a linear trend and gradually decreased. The change of asphalt penetration is −17.8%, −21.3%, −24.93%, and −28.96%, respectively, which indicates that the addition of TPU can increase the consistency and hardness of asphalt at the condition temperature, and thus improve the resistance of asphalt to high-temperature deformation [37]. After adding TPU, the softening point of asphalt gradually increases. When the TPU content increases from 0% to 20%, the softening point changes are +7.60%, +14.27%, +19.51%, and +19.71%, respectively, indicating that the addition of TPU can improve the high-temperature performance of asphalt [38]. This is in agreement with the conclusion obtained from the results of asphalt penetration test. However, with the increase in TPU added, the ductility value of asphalt showed a trend of increasing first and then decreasing, and the ductility change of asphalt was +38.05%, +16.81%, −6.19%, and −27.43%, respectively. When the TPU content was 15%, the ductility value of TPU-modified asphalt was 10.6 cm. It is slightly less than the ductility value of the base asphalt, indicating that excessive TPU will have a negative effect on the low-temperature performance of the asphalt [39]. This may be due to the fact that TPU, as a kind of high molecular polymer, may partially aggregate in asphalt, and the crosslinking structure of asphalt becomes sparse, which greatly weakens its flexibility and leads to its brittle characteristics at low temperatures.

The effect of SBR modifier content on the physical properties of asphalt is shown in Figure 8.

As can be seen from Figure 8, after adding a different dosage of SBR, the penetration value of asphalt showed a decreasing trend, and the change ranges were −2.3 mm, −2.9 mm, −3.4 mm, −4.5 mm, and −5.7 mm, respectively, while the softening point value of asphalt showed an increasing trend. The range values of SBR were +0.8 °C, +1.7 °C, +1.6 °C, +2.8 °C, and +3.1 °C, respectively. The changes in penetration and softening point were minimal, indicating that SBR provides limited improvement in the high-temperature performance of matrix asphalt. Additionally, pure SBR-modified asphalt demonstrates poor adhesion to aggregates and low compatibility with asphalt, leading to instability at elevated temperatures [40]. However, with increasing SBR content, there was a notable enhancement in asphalt ductility, with variation ranges of +37.4 cm, +37.8 cm, +51.4 cm, +53.0 cm, and +54.1 cm, respectively. This suggests that SBR-modified asphalt exhibits superior low-temperature performance [41]. This improvement may be attributed to the formation of silver lines resulting from the physical adsorption and swelling interactions between SBR and matrix asphalt. The development of these silver lines into a network structure helps maintain the integrity of the asphalt’s structural system, thereby enhancing its flexibility and ensuring that the asphalt retains excellent ductility in cold environments.

(2)Storage stability

The storage stability of modified asphalt with different TPU content was tested with a segregation test and the test results are shown in Table 7. As can be seen from Table 7, the softening point difference of TPU-modified asphalt is greater than that of matrix asphalt. With increasing TPU modifier content, the softening point of TPU-modified asphalt at the top and bottom gradually increases. Additionally, the softening point of the bottom of the asphalt is greater than that of the top, and the difference between the bottom and the top softening points gradually increases. The upper and lower softening point difference of the base asphalt is 0.2 °C, and after adding the TPU modifier, the upper and lower softening point difference of the modified asphalt gradually increases. When the content is 20%, the difference between the bottom softening point of the asphalt and the top softening point of the asphalt is the largest at 0.9 °C. This difference is still within the range required by the specification, indicating that the TPU-modified asphalt does not have a segregation phenomenon. The compatibility between TPU modifier and asphalt is good; that is, TPU-modified asphalt has excellent storage stability. The main reason is that the functional group of the TPU modifier reacts with some active groups in the asphalt, which increases the polarity of the asphalt material to a certain extent and reduces the polarity difference between the two components. This improves the compatibility between TPU and the matrix asphalt so that the TPU-modified asphalt shows good storage stability.

Table 8 shows the segregation test results of SBR-modified asphalt. With the increase of the content of the SBR modifier, the softening points of the bottom-modified asphalt are greater than that of the top-modified asphalt, and the difference of softening points between the bottom and the top tends to increase, with a maximum value of 1.8 °C. The main reason why the softening point difference increases with the increase of SBR content is that SBR is essentially synthetic rubber, which is different from asphalt in terms of solubility parameters, density. Even under the action of high-temperature and high-speed shear mixing, SBR is still difficult to be completely compatible with matrix asphalt, and phase separation may occur [42]. The softening point difference increases with the increase of SBR content, which means that the storage stability of SBR-modified asphalt is poor, but it still meets the specification requirements.

(3)Elastic recovery

Elastic recovery is a key technical indicator used to characterize the ability of asphalt to recover to its original state after being subjected to different degrees of external forces, which can intuitively reflect the durability and fatigue resistance of corresponding asphalt mixtures [43]. According to current research, the elastic recovery rate of modified asphalt is mainly used to characterize its elastic recovery ability. This article uses elastic recovery tests to study the variation of asphalt elastic recovery ability with the dosage of TPU and SBR modifiers. The test results are shown in Figure 9 and Figure 10, respectively.

As can be seen from Figure 9, the content of the TPU modifier has a significant impact on the elastic recovery performance of modified asphalt. With the increase of TPU content, the elastic recovery rate of modified asphalt presents a linear trend of increasing the elastic recovery rates at TPU dosages of 5%, 10%, 15%, and 20% are 47%, 53.4%, 58.7%, and 63.9%, respectively. Therefore, the elastic recovery performance is enhanced, indicating that the viscoelastic ratio of asphalt changes after TPU is added, and that the elastic component ratio of asphalt gradually increases with the increase of the content. Thus, the elastic recovery ability of TPU-modified asphalt is effectively improved. Because TPU powder can significantly improve the modulus of asphalt and form a continuous phase with asphalt, which acts as a frame for matrix asphalt and restrains the flow deformation of asphalt. When asphalt is subjected to external forces, polyurethane formed in a continuous phase can absorb most of the stress and improve the resistance to deformation of asphalt. It also significantly improves the elastic properties of asphalt [44].

It can be seen from Figure 10 that the elastic recovery rate of the modified asphalt increases after the incorporation of SBR, indicating that the elastic recovery performance of the SBR-modified asphalt is gradually improved. When the dosage is greater than 3.5%, the improvement rate of elastic recovery starts to slow down, and when the dosage is 4%, the elastic recovery rate reaches the maximum (38.9%). This is due to the fact that after a certain amount of SBR modifier is added to the matrix asphalt, the molecular structure of SBR can absorb some light components in the matrix asphalt when SBR reacts physically with the matrix asphalt, changing the internal composition structure of the asphalt, and improving the elastic recovery ability of SBR-modified asphalt to a certain extent. However, due to the weak elasticity of SBR itself, and when the SBR content reaches a certain degree, the light components in the matrix asphalt have been completely adsorbed by SBR, and the elastic recovery ability of asphalt cannot be further improved by adding more SBR.

To sum up, in order to fully guarantee the high and low-temperature performance, storage stability, and elastic recovery performance of TPU-modified asphalt and SBR-modified asphalt, the optimal dosage of TPU modifier and SBR modifier is 15% and 3.5%, respectively.

#### 3.1.2. Performance Analysis of Composite-Modified Asphalt

According to Section 3.1.1, the basic properties of TPU-modified asphalt and SBR-modified asphalt are comprehensively analyzed. It is known that the optimal content of the TPU modifier is 15%, and the optimal content of the SBR modifier is 3.5%. In this subsection, four different types of TPU/SBR composite-modified asphalt were prepared with TPU modifier (10% and 15%) and SBR modifier (3.0% and 3.5%): 3.0% SBR+10% TPU (3S/10T), 3.0% SBR+15% TPU (3S/15T), 3.5% SBR+10% TPU (3.5S/10T), and 3.5% SBR+15% TPU (3.5S/15T).

(1)Physical properties

The effects of TPU and SBR content on the physical properties of the composite-modified asphalt are shown in Figure 11.

As can be seen from the figure, when the content of TPU/SBR composite modifier increases, the injection degree of composite-modified asphalt significantly decreases, which indicates that the composite modifier can effectively increase the consistency of asphalt, and then improve its stability at high temperatures. When the TPU content is constant, the injection degree of asphalt is not significantly reduced when the SBR content is increased. When the TPU content is 15%, the injection degree of composite-modified asphalt with different SBR content is decreased by 0.7 m. However, when the SBR content is the same, the injection degree of asphalt is significantly decreased when the TPU content is increased. The injection degree of modified asphalt with different TPU content is reduced by 4.5 m, which indicates that SBR has a limited effect on improving the high-temperature stability of asphalt, while TPU can significantly improve the high-temperature stability of asphalt. At the same time, the softening point of the composite-modified asphalt increases with the increase of the composite modifier. When the TPU content is constant, the softening point of the asphalt increases with the increase of SBR content. For example, when the TPU content is 10%, the softening point of the composite-modified asphalt with different SBR content increases by 0.4 °C. However, when the SBR content remained unchanged, the softening point of asphalt increased significantly with the increase of TPU content. When the SBR content was 3.5%, the softening point of composite-modified asphalt with different TPU content increased by 2.3 °C, indicating that the improvement effect of TPU on the high-temperature stability of composite-modified asphalt was significantly better than that of SBR. The reason for this phenomenon may be that the compatibility between SBR and asphalt is not very good, and the products after the reaction of TPU with matrix asphalt affect the formation of micellar between SBR and asphaltene in asphalt at the same time, so the improvement of needle penetration and softening point of asphalt by TPU is better than that by SBR.

In addition, it can be seen from Figure 11 that in composite-modified asphalt, SBR plays a leading role in improving the low-temperature performance of asphalt, and the improvement effect is better with the increase of SBR content. When the TPU content is unchanged, the ductility of the composite-modified asphalt increases significantly with the increase of the SBR content. When the TPU content is 15%, the ductility value of the composite-modified asphalt with the SBR content is 6.7 cm higher than that of the SBR content is 3%, but when the SBR content is constant, the ductility of the asphalt decreases with the increase of the TPU content. When the SBR content is 3.5%, the ductility value of the composite-modified asphalt with 15% TPU content is 1.8 cm lower than that of the composite-modified asphalt with 10% TPU content; that is, the ductility improvement effect of the composite-modified asphalt is better when the TPU content is smaller. All these indicate that the presence of SBR can significantly improve the low-temperature performance of the composite-modified asphalt. However, TPU will have a certain weakening effect on the low-temperature crack resistance of asphalt

(2)Storage stability

The absence of segregation in polymer-modified asphalt is a prerequisite for maintaining its original excellent performance. Testing the separation softening point difference can to some extent reflect the ability of the modifier to disperse and fuse with asphalt during the compatibility process, as well as the stability of each component after blending [45]. The segregation test results of TPU/SBR composite-modified asphalt are shown in Figure 12.

As can be seen from Figure 12, the softening point difference of TPU/SBR composite-modified asphalt shows an increasing change rule with the increase of composite modifier, but the change is not very obvious, and the maximum value is only 1.3 °C, which is still within the range required by the specification, indicating that composite-modified asphalt generally does not produce segregation and delamination. In TPU/SBR composite-modified asphalt, when the TPU content is the same, the SBR content increases and the softening point difference increases. When the TPU content is 10%, the softening point difference of the composite-modified asphalt with different SBR content increases by 0.2 °C. The reason why the softening point difference value increases with the increase of SBR content may be that SBR, as a synthetic rubber, has certain differences in solubility parameters and density components with asphalt. Even after high-temperature and high-speed shear mixing and dispersion, SBR makes it difficult to achieve complete compatibility with the base asphalt and may even lead to partial aggregation of heavy components in the asphalt. At the same time, the polarity characteristics of SBR-modified asphalt will also change accordingly, resulting in an increase in softening point difference value with the increase of SBR content, which also means that the storage stability of SBR-modified asphalt is poor. When the SBR content is the same, the larger the TPU content is, the smaller the softening point difference is. When the SBR content is 3%, the softening point difference of composite-modified asphalt with different TPU content is reduced by 0.1 °C which is still within the scope of regulatory requirements. It indicates that TPU-modified asphalt has not undergone segregation, and the compatibility between TPU modifier and asphalt is good; that is, TPU-modified asphalt has excellent storage stability. Related studies have shown that the storage stability of different materials mainly depends on the polarity difference between them. The larger the polarity difference, the worse the storage stability, and the smaller the polarity difference, the better the storage stability [46]. TPU modifier does not undergo condensation during high-temperature storage, minimizing the migration and segregation of the modifier. The main reason for this is the reaction between the -NCO groups in TPU and the active hydrogen atoms (mainly -OH) in asphaltene micelles [47], which reduces the polarity difference between material components and effectively improves the compatibility between TPU and matrix asphalt. Based on the change of softening point difference, it can be preliminarily concluded that the presence of TPU can improve the compatibility between SBR and asphalt. The possible segregation phenomenon of SBR-modified asphalt is alleviated, resulting in the achievement of reducing the segregation degree of complex modified asphalt systems and improving the overall storage stability. The segregation test is to analyze the segregation degree of polymer-modified asphalt from a macro perspective, and the test results may have some errors, which cannot accurately reflect the compatibility of the polymer modifier and asphalt. Subsequently, the storage stability of TPU/SBR composite-modified asphalt will be further discussed from a microscopic perspective by fluorescence microscopy.

(3)Elastic recovery

The effect of TPU/SBR composite modifier on the elastic recovery performance of asphalt is shown in Figure 13 and Figure 14, respectively.

The phase angle δ reflects the changes in stress and strain of asphalt under external loads and can also reflect the proportional relationship between the two components of asphalt viscosity and elasticity. The larger the phase angle δ value, the more viscous components there are in asphalt than elastic components. To some extent, this improves the flow and deformation ability of asphalt, while weakening the deformation recovery ability, which is then manifested as asphalt pavement being more prone to rutting diseases [48]. The temperature scanning test results of 7 different types of asphalt are shown in Figure 13. It can be seen that the phase angle δ of asphalt is positively correlated with temperature; that is, as the temperature increases, the phase angle δ and the viscosity component also increase. Asphalt is more prone to softening under high-temperature conditions, leading to permanent deformation. Under the same temperature conditions, the phase angle δ of the matrix asphalt is the largest, showing a size pattern of 70 # > 3.5% SBR > 3% SBR+10% TPU > 3.5% SBR+10% TPU > 15% TPU > 3% SBR+15% TPU > 3.5% SBR+15% TPU. This indicates that the addition of modifiers causes significant changes in the viscoelastic composition of asphalt, with a decrease in viscous components and an increase in elastic components, thereby improving the elastic recovery ability of asphalt and effectively improving high-temperature rutting resistance. In addition, at the same temperature, the phase angle δ of the four types of composite-modified asphalt is smaller than that of SBR-modified asphalt. Among the composite-modified asphalt, the TPU content has the most significant effect on its phase angle δ, indicating that the addition of TPU to SBR-modified asphalt effectively reduces the proportion of viscous components and increases the proportion of elastic components, thereby enhancing the high-temperature deformation resistance of asphalt. When TPU and SBR are used for composite modification of matrix asphalt, the interaction between the two modifiers significantly enhances the deformation recovery ability and high-temperature rutting resistance of the composite-modified asphalt.

According to Figure 14, it can be seen that the elastic recovery rate of TPU/SBR composite-modified asphalt increases with the increase of the content of TPU and SBR, and TPU is superior to SBR in improving the elastic recovery performance. When the content of TPU is 15%, the elastic recovery rate of the modified asphalt with 3.5% SBR is 5.8% higher than that with 3% SBR. When the content of SBR is 3.5%, the elastic recovery rate of the compound-modified asphalt with 15% TPU content increases by 11.3% compared with that with 10% TPU content; also, the growth rate is faster. The results showed that TPU/SBR composite modifier could significantly enhance the overall elastic recovery ability of the composite-modified asphalt, and TPU was the dominant one. The reason may be that during the physical reaction between SBR and asphalt, heavy components with greater polarity in the asphalt will adsorb the light components, and the phenomenon of agglomeration and uneven dispersion easily occurs when the heavy components are relatively increased, while the crosslinking effect of light components in the system after the reaction between TPU and asphalt is enhanced, so that the asphaltenes and colloid in the asphalt can be more evenly dispersed in the system. The problem of asphalt composition agglomeration is effectively solved, and a relatively stable cross-linked network structure is formed inside the composite-modified asphalt system which enhances the elasticity of the asphalt system [49].

In summary, TPU can enhance the compatibility between SBR and asphalt, solve the problem of local agglomeration of SBR-modified asphalt components, improve the overall elasticity of composite-modified asphalt, and maintain good performance of composite-modified asphalt under high-temperature conditions.

### 3.2. Analysis of Micro-Properties

#### 3.2.1. Atomic Force Microscopy (AFM) Test

In order to study the micro-morphology of TPU/SBR composite-modified asphalt, AFM tests were performed on four different types of asphalt as-is samples (including base asphalt, 3.5%SBR, 15%TPU, 3.5%SBR+15%TPU).

The structure of the asphalt micro-surface is divided into two types: continuous phases and dispersed phases. The continuous phase is composed of light components such as aromatic components and saturated components. The dispersed phase is a honeycomb structure formed by the colloid in the heavy component wrapping the asphaltene, which is dispersed in the continuous phase [50,51].

Figure 15 shows the 2D and 3D morphologies of 70# matrix asphalt. From the two-dimensional morphology, it can be seen that the asphalt surface is composed of alternating light and dark similar to the shape of bees, which is called “bee structure” according to its shape characteristics. Related studies have found that the “bee structure” is composed of two molecular components with large polarity differences in asphalt. In the chemical composition of asphalt, the heavy component composed of asphaltene and gum has a greater polarity, thus forming a brighter area in the “bee structure”, while the light component composed of aromatic components and saturated compounds has less polarity and is adsorbed around the asphaltene with greater polarity. In addition, the saturated substance containing long alkyl side chains is relatively small in molecular size and can be interspersed between asphaltene and gum, forming a “bee structure” subsidence area, which is denoted as the darker color area in the topography [52,53]. In addition, other scholars have found that due to the different thermal conductivity between asphaltene and saturated material, there are great differences in their mechanical properties such as shrinkage and deformation ability under different temperature conditions. Therefore, the asphalt surface will fluctuate and change, forming a shape similar to a “fold”. The “bee structure” appears on the two-dimensional morphology pattern [54]. These studies all show that the formation of “bee structure” is closely related to the interaction between chemical components and molecules in asphalt.

It can be seen from the three-dimensional microscopic morphology of asphalt that the length, height, and size of these “bee structures” are not exactly the same. According to the structural characteristics, the more typical structures in the three-dimensional morphology map are classified and labeled accordingly. No. 1 represents a large “bee structure”, No. 2 represents a medium-sized “bee structure”, and No. 3 represents a small “bee structure”. Among them, the convex part of No. 1 is relatively high, long, and bright, indicating that asphaltene is concentrated in this area. The “bee structure” is lighter in color, while the surrounding part is darker in color, indicating that the heavy component is dispersed in the light component, and the two complement each other and interact, making the asphalt reflect certain mechanical properties.

The AFM test results of SBR-modified asphalt are shown in Figure 16. According to the analysis of 2D and 3D micro-topography, there are also “bee structures” of different sizes in SBR-modified asphalt. These “bee structures” with alternating light and dark are more obvious, and the local area increases, which may be due to the local aggregation of asphaltene and other heavy components in the asphalt during the physical reaction of adsorption and swelling between SBR and matrix asphalt. Moreover, the polar property difference with saturated grade light components increases, so that the local “bee structure” in the asphalt is more significant, the distribution of these structures is not uniform, and there is a local aggregation phenomenon. Therefore, the high-temperature performance and storage stability of SBR-modified asphalt are poor.

Figure 17 shows the 2D and 3D morphologies of TPU-modified asphalt. Compared with matrix asphalt, TPU-modified asphalt has a smaller bee shape structure, shorter height and size, more quantity, and more uniform distribution, indicating that the addition of TPU modifier can make the dispersed phase composed of heavy components dissolve well in the continuous phase, which can again verify that TPU modifier can stably combine with the matrix asphalt, have good compatibility, and form a more stable system. Therefore, it has a higher softening point and lower penetration, which can improve the high-temperature performance of asphalt.

Figure 18 shows the 2D and 3D morphologies of TPU/SBR composite-modified asphalt. A comparative analysis of the microstructure of matrix asphalt and TPU/SBR composite-modified asphalt shows that with the increase of TPU content, the height size of “bee structure” decreases and the distribution becomes more uniform, indicating that the addition of TPU can reduce the polarity difference between the asphaltene in matrix asphalt, SBR-modified asphalt, and its nearby chemical components, and effectively enhance the compatibility between different components. In addition, asphaltene and gum can be well dispersed to the surrounding aromatic fractions and saturated substances, and these chemical components cross-interact with each other to form a more stable cross-linked network structure system of composite-modified asphalt, which makes it difficult for the internal molecules of TPU/SBR composite-modified asphalt to move relative under high-temperature conditions and external loads. It shows excellent resistance to deformation at high temperatures.

#### 3.2.2. Fluorescence Microscope (FM) Test

In this paper, the FM test of 5 different bitumen was carried out with IMAGER.Z2 fluorescence microscope (from Carl Zeiss Optics GMBH, Germany). At present, the main methods for making FM test samples are the mold test method, hot drop cover slide method, and freeze forming fold section method. The hot drop cover slide method is relatively simple to operate and has no special requirements on the test environment temperature, so this paper adopts the hot drop cover slide method for sample preparation. The observation magnification of the sample was 10 times.

Figure 19a is the fluorescence diagram of the matrix asphalt, which shows a single background color and no fluorescence points, indicating that the matrix asphalt has a single-phase structure. Figure 19b–e are all fluorescence diagrams of modified asphalt, in which there are scattered fluorescent points of different sizes, indicating that the modifier will produce corresponding fluorescence phenomenon under ultraviolet irradiation of fluorescence microscope, and the phase structure of asphalt will be transformed from a single phase to a dispersed phase.

Figure 19b is the fluorescence diagram of SBR-modified asphalt, from which it can be seen that the irregular distribution of fluorescent dots of different sizes. In the figure, some fluorescent dots showed aggregation phenomenon, and the fluorescence phenomenon was obvious, indicating that the addition of SBR modifier was not completely compatible with the matrix asphalt, and the storage stability of SBR-modified asphalt was relatively poor on a macro level. It should be noted that the shape of individual fluorescent dots in the figure is relatively large, which may be caused by the uneven shear during the preparation of SBR-modified asphalt.

Figure 19c shows the fluorescence diagram of TPU-modified asphalt. Compared with SBR-modified asphalt, the fluorescence points of TPU-modified asphalt are less and more evenly distributed, indicating that TPU can be effectively integrated into the asphalt system and converted into the internal components of asphalt. This is mainly because after the reaction of TPU with the base asphalt, the polarity difference between the heavy and light components in the modified asphalt is reduced, so as to effectively improve the compatibility between TPU and the base asphalt.

Figure 19d,e show the fluorescence diagrams of 3.5%SBR+10%TPU composite-modified asphalt and 3.5%SBR+15%TPU composite-modified asphalt, respectively. It can be seen from the two figures that when the content of SBR is constant, the fluorescence point of the composite-modified asphalt decreases with the increase of TPU content. In addition, compared with TPU-modified asphalt, the fluorescence diagram of composite-modified asphalt showed an increase in fluorescent dots. Conversely, compared with SBR-modified asphalt, the fluorescence diagram of composite-modified asphalt shows a decrease in fluorescent dots, with a more uniform distribution and no obvious agglomeration. This indicates good compatibility between TPU, SBR, and matrix asphalt. TPU/SBR composite-modified asphalt has a good phase structure, which also explains the reason why the softening point difference decreases with the increase of TPU content in the segregation test of composite-modified asphalt. TPU can effectively reduce the possible segregation phenomenon of SBR-modified asphalt, thus improving the overall storage stability of TPU/SBR composite-modified asphalt.

#### 3.2.3. Fourier Transform Infrared Spectroscopy (FTIR) Test

In order to study the modification mechanism of TPU/SBR composite-modified asphalt, seven different types of original asphalt (including 70# base leach (B), 3.5%SBR (3.5S), 15%TPU (15T), 3.0%SBR+10%TPU (3S/10T), 3.0%SBR+15%TPU (3S/15T), 3.5%SBR+10%TPU (3.5S/10T), 3.5%SBR+15%TPU (3.5S/15T)) were tested by Fourier transform infrared spectroscopy (FTIR).

The FT-IR test results of matrix asphalt, SBR-modified asphalt, and TPU-modified asphalt are shown in Figure 20.

By comparing the infrared spectral curves of matrix asphalt and SBR-modified asphalt, it can be seen that there is no significant difference between the curves of the two kinds of asphalt; that is, there is no new characteristic peak of SBR-modified asphalt, indicating that after adding SBR modifier to the matrix asphalt, the modified asphalt does not generate new functional groups, SBR and asphalt do not react chemically, and the interaction between the two is mainly physical fusion. It can be inferred that the main reason for the change in the properties of SBR-modified asphalt compared with the matrix asphalt is the change in the composition and molecular arrangement of the asphalt after the incorporation of SBR.

In addition, through comparative analysis of the infrared spectral curves of matrix asphalt and TPU-modified asphalt, it can be seen that there are obvious differences in the characteristic absorption peaks of the two types of asphalt and that TPU-modified asphalt has new characteristic absorption peaks at 1730 cm^−1^, 1191 cm^−1^, and 1177 cm^−1^. At the same time, there are some weak characteristic absorption peaks in the wavelength range of 1160~1000 cm^−1^. The peak position of 1730 cm^−1^ represents the stretching vibration absorption peak of the C=O group in the carbamate or urea group, which is mainly due to the chemical reaction between the -NCO group in polyurethane and the polar groups containing H, O, and N atoms in asphalt. Then, carbamate (-NHCO) or urea ($H_{2}{\mathrm{NCONH}}_{2}$) structures containing carbonyl C=O groups are formed [55,56]. The absorption peaks in the range of 1191 cm^−1^, 1177 cm^−1^, and 1160–1000 cm^−1^ are located in the fingerprint region, among which the first two are the expansion vibration absorption peaks of tertiary alcohols, and the vibration absorption peaks of the S=O group in sulfoxide group and ether-O group in aminoester group are mainly found in the 1160–1000 cm^−1^ region. It can be speculated that after adding a TPU modifier to asphalt, chemical reactions can occur between the -NCO group in TPU and some polar groups in asphalt (such as OH), forming a carbamate structure. In other words, TPU chemically modifies asphalt to form a cross-linked network structure, which significantly enhances the interaction force between molecules in asphalt. This explains the good storage stability and high-temperature performance of TPU-modified asphalt.

The FT-IR test results of matrix asphalt, 3.5%SBR+10%TPU composite-modified asphalt, and 3.5%SBR+15%TPU composite-modified asphalt are shown in Figure 21.

Compared with matrix asphalt and SBR-modified asphalt, TPU/SBR composite-modified asphalt has a more obvious characteristic absorption peak at 1730 cm^−1^, and a weaker characteristic peak change in the region of 1305~1030 cm^−1^, which is not much different from the characteristic absorption peak of TPU-modified asphalt, but the degree of change is slightly different. When the content of SBR is constant, the characteristic absorption peak strength of composite-modified asphalt increases with the increase of TPU content. This shows that the presence of the SBR modifier will slightly interfere with the chemical reaction between TPU and matrix asphalt in the composite-modified asphalt, but with the increase of TPU content, this interference effect is gradually weakened, or even negligible.

Based on the above analysis, SBR is mainly modified by physical reaction with matrix asphalt, and physical reinforcement is carried out by adsorption of molecular components in matrix asphalt and swelling development, thus effectively improving the low-temperature performance of asphalt. The chemical reaction between the -NCO group of TPU and the polar group (-OH) of the matrix asphalt is obvious, the structure of carbamate is formed, and the interaction force between the asphalt molecules is significantly enhanced. TPU/SBR composite modifier can physically and chemically modify the matrix asphalt at the same time, and the compatibility between TPU and SBR is good; therefore, TPU/SBR composite-modified asphalt has excellent performance of the two kinds of modified asphalt. Thus, TPU/SBR composite-modified asphalt shows excellent high-temperature stability and low-temperature crack resistance at the same time.

## 4. Conclusions

This study investigates the modification of 70# matrix asphalt using thermoplastic polyurethane (TPU) and styrene-butadiene rubber (SBR) as modifiers. Initially, routine performance tests were conducted to assess the physical properties, storage stability, and elastic recovery of TPU-modified and SBR-modified asphalts at various dosages. Based on these technical performance indicators, the optimal dosages for both modifiers were determined. Subsequently, TPU/SBR composite-modified asphalt was prepared, and its performance changes were analyzed. Using atomic force microscopy, fluorescence microscopy, and Fourier transform infrared spectroscopy, the microstructure, phase structure, and modification mechanisms of the asphalt before and after modification were examined, with an in-depth analysis of the underlying reasons for performance changes. The main conclusions are as follows:

(1) TPU increases the viscosity of asphalt, with a more pronounced effect at higher dosages. It exhibits good compatibility with asphalt, leading to enhanced high-temperature performance and storage stability. However, ductility slightly decreases, and low-temperature performance is somewhat reduced. Based on conventional performance, a recommended TPU dosage is 15%.

(2) SBR significantly enhances the flexibility of asphalt, thereby improving its low-temperature performance. However, as the dosage increases, compatibility with asphalt decreases, resulting in poor storage stability. Based on conventional performance, a recommended SBR dosage is 3.5%.

(3) The addition of TPU and SBR improves the conventional performance of asphalt. Compared to TPU-modified asphalt, the ductility of the composite-modified asphalt is significantly increased, indicating that SBR enhances the low-temperature ductility of TPU-modified asphalt. TPU improves the viscosity and compatibility of SBR-modified asphalt, thus enhancing its high-temperature performance and storage stability.

(4) TPU facilitates the even dispersion of the heavy components in SBR-modified asphalt into lighter components, improving its structural composition. It forms a more stable cross-linked network structure, effectively constraining molecular movement under high-temperature conditions and enhancing high-temperature deformation resistance.

(5) Matrix asphalt initially has a single-phase structure. The addition of TPU and SBR transforms it into a dispersed phase structure. The good compatibility among the components results in TPU/SBR composite-modified asphalt exhibiting excellent storage stability.

(6) SBR undergoes a physical reaction with asphalt, significantly enhancing the low-temperature performance of asphalt through physical blending. TPU-modified asphalt exhibits distinct new characteristic absorption peaks at 1730 cm^−1^, 1191 cm^−1^, and 1177 cm^−1^, indicating a chemical reaction between the -NCO group of TPU and certain polar groups of asphalt (such as -OH). Therefore, the composite-modified asphalt system undergoes both physical and chemical modifications, resulting in excellent comprehensive performance.

## 5. The Limitations and Future Scope of the Study

This study systematically investigates the conventional properties and microscopic characteristics of TPU/SBR composite-modified asphalt. However, due to limitations in my academic capacity, a comprehensive analysis of the properties of TPU/SBR composite-modified asphalt remains incomplete. It is recommended that future research address the following areas:

(1) The impact of various preparation processes on asphalt performance, including but not limited to shear temperature, rate, and duration. Variations in these factors may lead to significant changes in asphalt performance, warranting more detailed investigations.

(2) An exploration of the effects of thermal oxidation and ultraviolet aging on the performance of TPU/SBR composite-modified asphalt. Additionally, research should examine the performance changes of TPU/SBR composite-modified asphalt and its mixtures before and after aging.

## Figures and Tables

**Figure 1 polymers-16-02766-f001:**
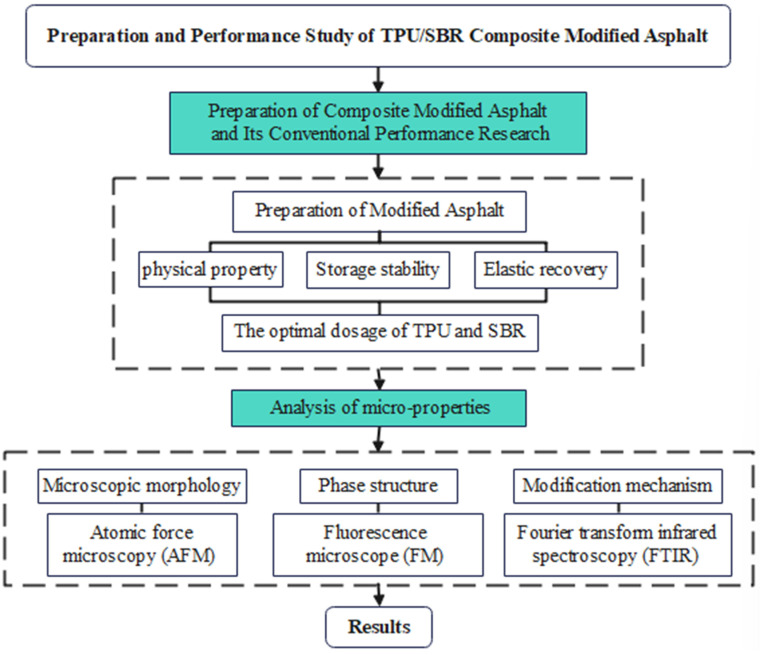
Technology roadmap.

**Figure 2 polymers-16-02766-f002:**
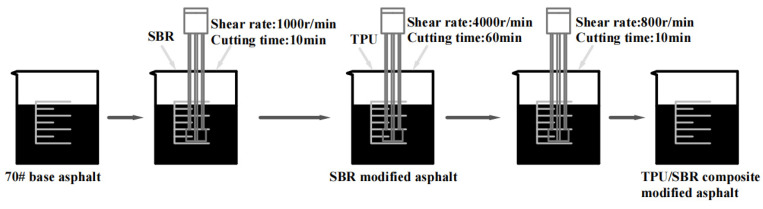
The preparation process of composite-modified asphalt.

**Figure 3 polymers-16-02766-f003:**
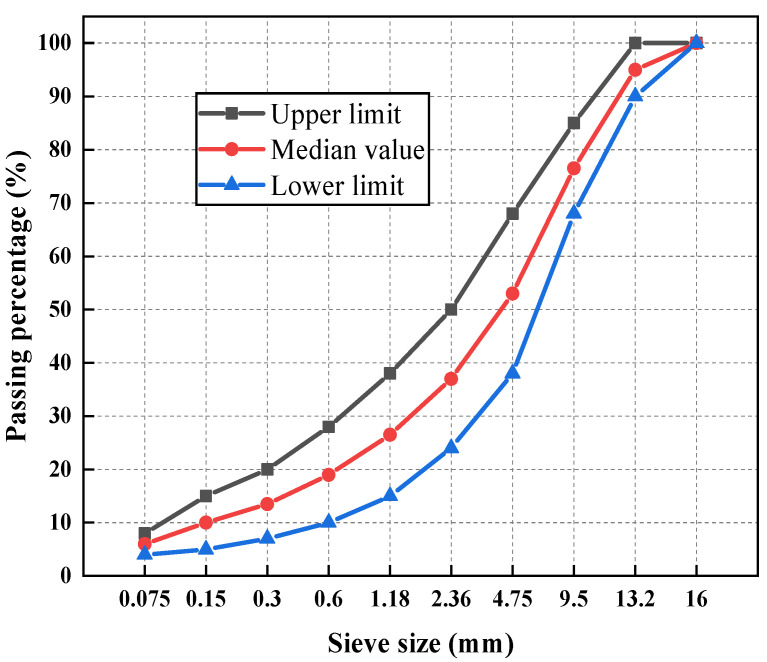
Grading design curve of asphalt mixture.

**Figure 4 polymers-16-02766-f004:**
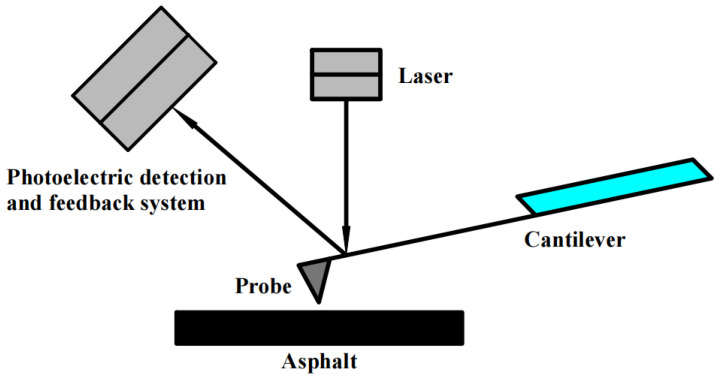
Test principle of AFM.

**Figure 5 polymers-16-02766-f005:**
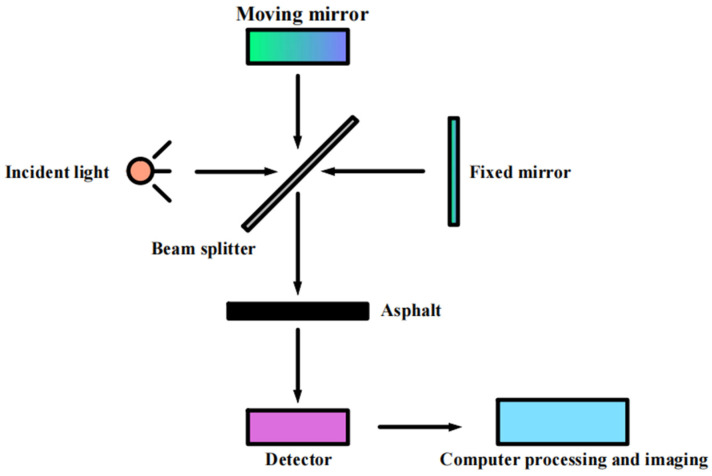
Test principle of FT-IR.

**Figure 6 polymers-16-02766-f006:**
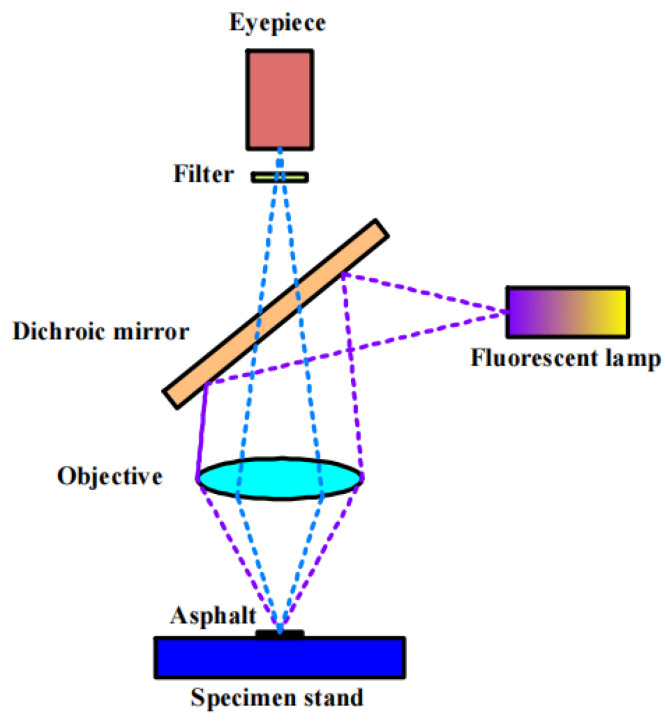
Test principle of FM.

**Figure 7 polymers-16-02766-f007:**
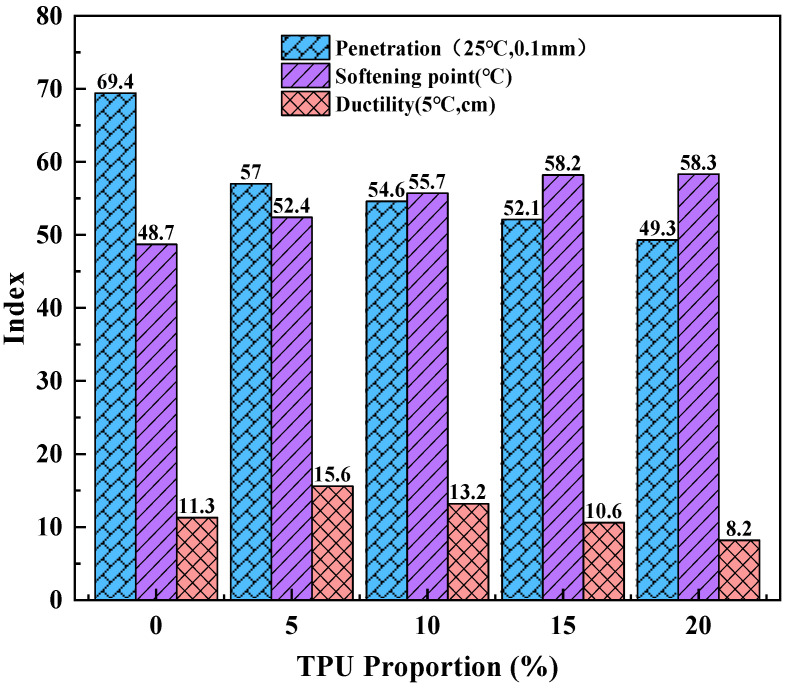
Physical properties of TPU-modified asphalt.

**Figure 8 polymers-16-02766-f008:**
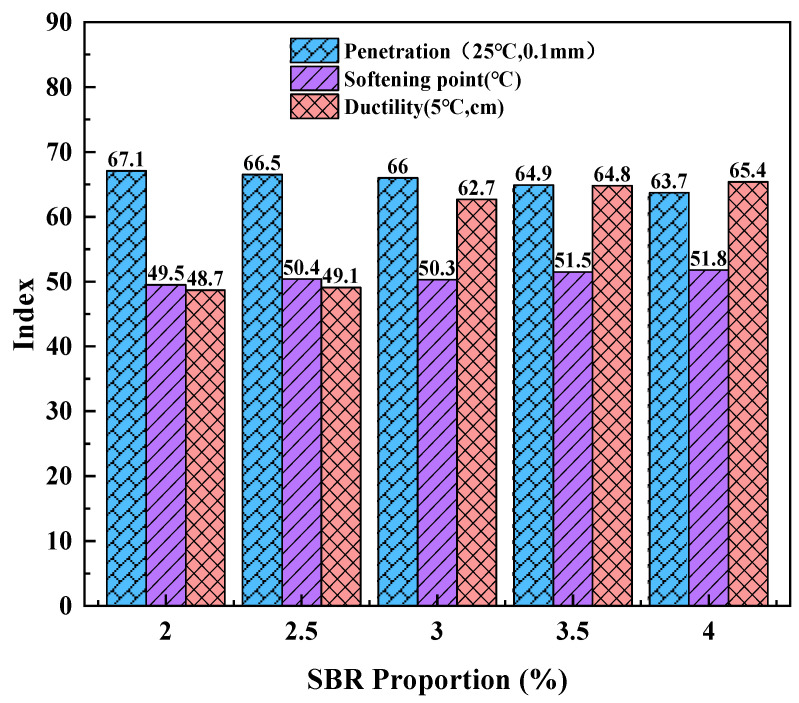
Physical properties of SBR-modified asphalt.

**Figure 9 polymers-16-02766-f009:**
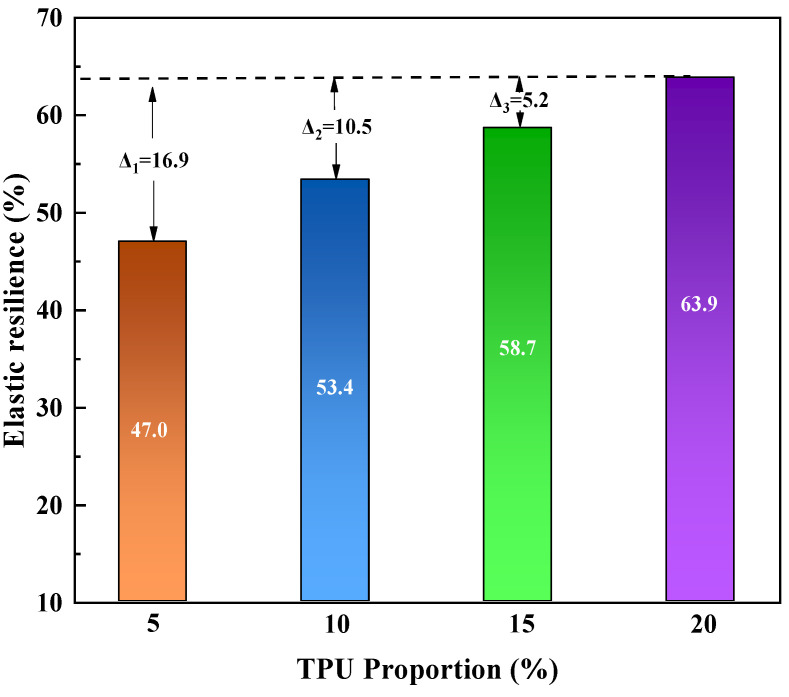
Elastic recovery rate of TPU-modified asphalt.

**Figure 10 polymers-16-02766-f010:**
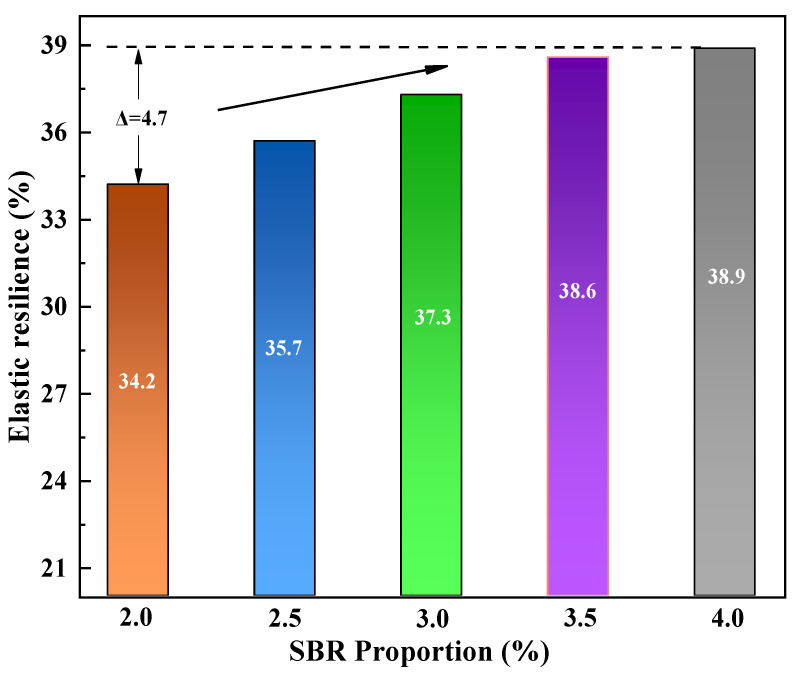
Elastic recovery rate of SBR-modified asphalt.

**Figure 11 polymers-16-02766-f011:**
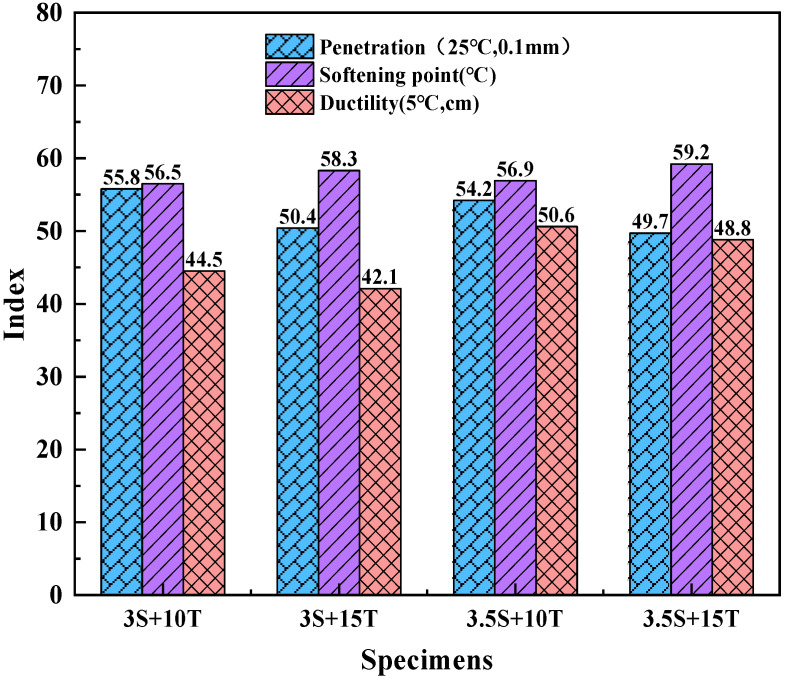
Physical properties of TPU/SBR composite-modified asphalt.

**Figure 12 polymers-16-02766-f012:**
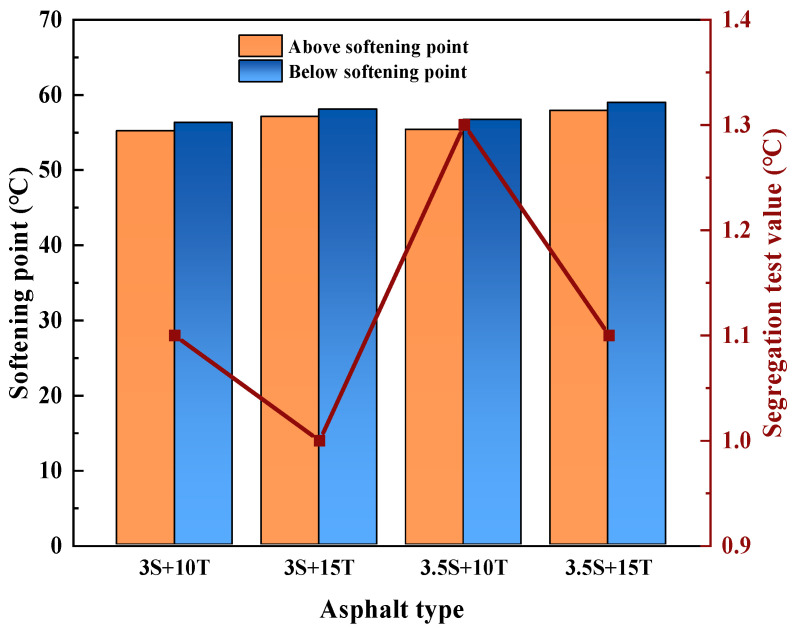
Differences in softening points of different composite-modified asphalt segregation tests.

**Figure 13 polymers-16-02766-f013:**
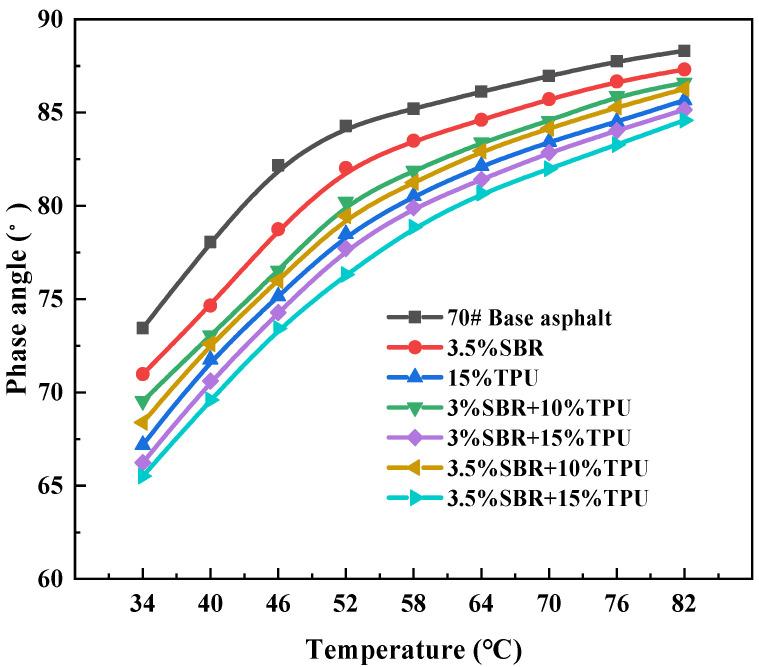
Curve of asphalt phase angle with temperature.

**Figure 14 polymers-16-02766-f014:**
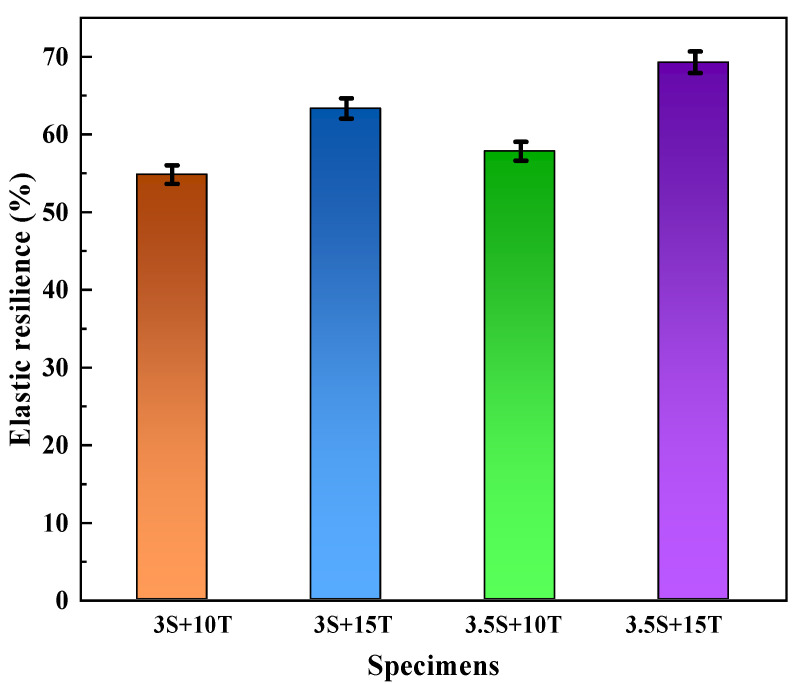
Elastic recovery rate of different composite-modified asphalt.

**Figure 15 polymers-16-02766-f015:**
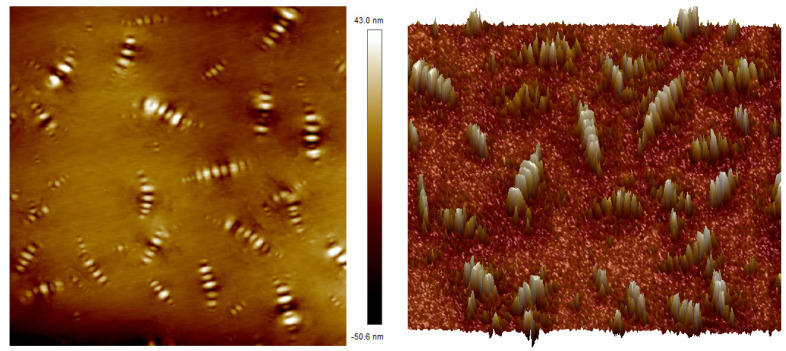
Micro-morphology of base asphalt (left: 2D; right: 3D).

**Figure 16 polymers-16-02766-f016:**
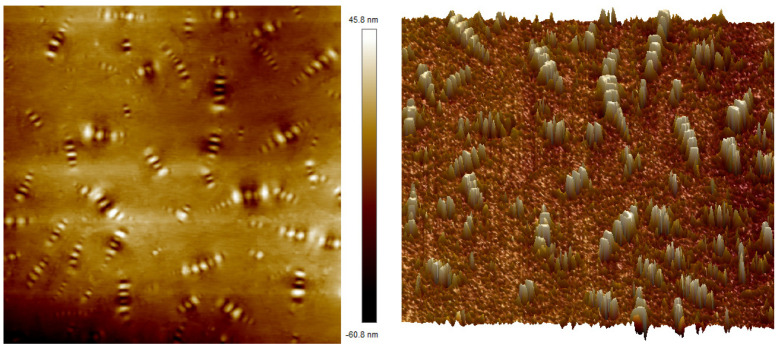
Micro-morphology of SBR-modified asphalt (**left**: 2D; **right**: 3D).

**Figure 17 polymers-16-02766-f017:**
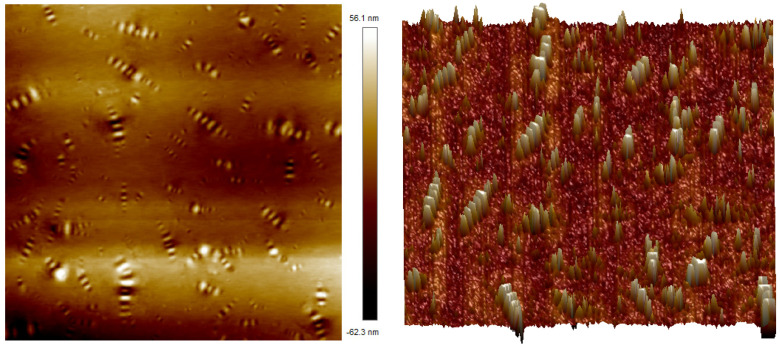
Micro-morphology of TPU-modified asphalt (**left**: 2D; **right**: 3D).

**Figure 18 polymers-16-02766-f018:**
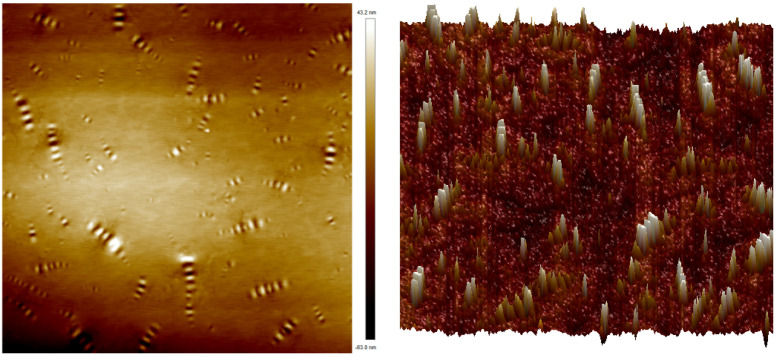
TPU/SBR composite-modified asphalt micro-morphology (**left**: 2D; **right**: 3D).

**Figure 19 polymers-16-02766-f019:**
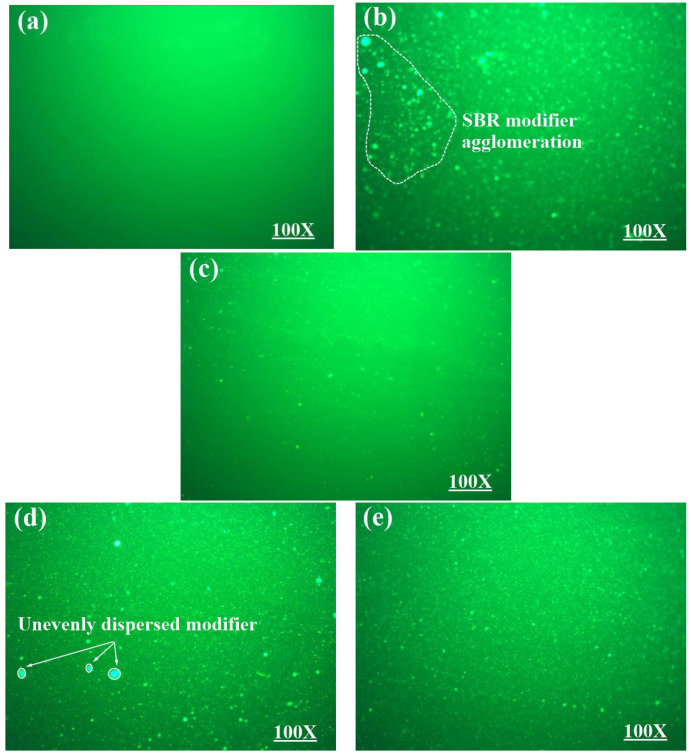
FM test results of different asphalt: (**a**) 70#; (**b**) SBR; (**c**) TPU; (**d**) 3.5S+10T; (**e**) 3.5S+15T.

**Figure 20 polymers-16-02766-f020:**
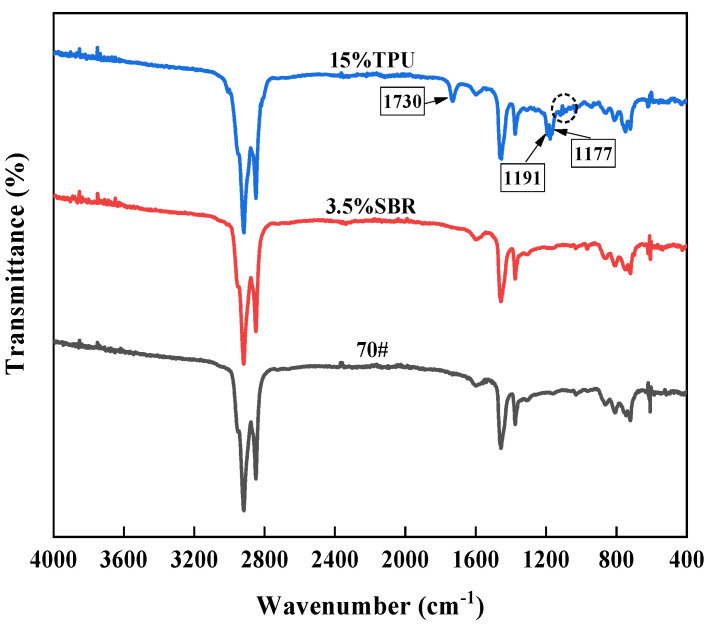
FTIR of single mixed modified asphalt.

**Figure 21 polymers-16-02766-f021:**
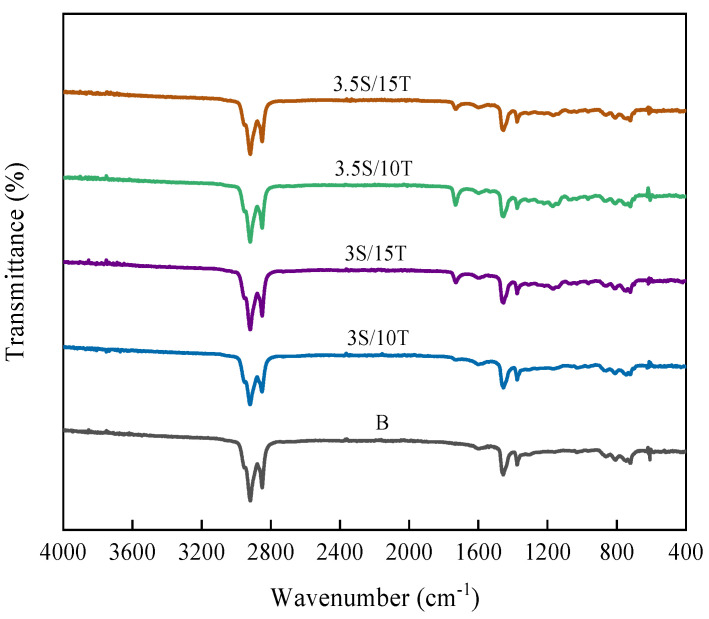
FTIR of composite-modified asphalt.

**Table 1 polymers-16-02766-t001:** Technical property indexes of 70# base asphalt.

Test Parameters		Units	Technical Standards	Test Results
Penetration (25 °C, 5 s, 100 g)		0.1 mm	60~80	69.4
Penetration Index (PI)		—	−1.5~+1.0	−0.88
Softening Point (Ring-and-Ball Method, 5 °C)		°C	≥46	48.7
Rotational Viscosity (135 °C)		mPa·s	<3000	475
Ductility (cm)	5 °C (5 cm/min)	cm	measured	11.3
	15 °C (5 cm/min)	cm	≥100	>100
Wax Content (Distillation Method,%)		%	≤2.2	0.85
Flash Point (Open Cup)		°C	≥260	>300
Solubility (Trichloroethylene)		%	≥99.5	99.86
Density (15 °C)		g/cm^3^	measured	1.036
RTFOT Test	Mass Change	%	±0.8	−0.15
Residue	Penetration Ratio (25 °C)	%	≥61	74.0
(163 °C, 5 h)	Residual Ductility (10 °C)	%	≥6	6.5

**Table 2 polymers-16-02766-t002:** Properties of TPU.

Test Parameters	Units	Test Results
Appearance	—	White Particles
Particle Size	Mesh	32
Density	g/cm^3^	1.11
Tensile Strength	MPa	55.3
Elongation at Break	%	479.1

**Table 3 polymers-16-02766-t003:** Basic properties of SBR.

Test Parameters	Units	Technical Standards	Test Results
Appearance	—	White to slightly yellow powder	White to slightly yellow powder
Molecular Weight	MW	20~30	28
Particle Size	Mesh	Approximately 20	20
Styrene Content	%	22.5~24.5	23.5
Mooney Viscosity	Pa·s	50~70	67
Moisture Content	%	≤25	1.6

**Table 4 polymers-16-02766-t004:** Technical indexes of coarse aggregate.

Test Parameters	Units	Technical Standards	Test Results
Aggregate Crushing Value	%	≤26	12.6
Apparent Relative Density	—	≥2.60	3.1
Bulk Relative Density	—	—	2.967
Los Angeles Abrasion Loss	%	≤28	11.3
Flakiness and Elongation Index	%	≤15	9.4
<0.075 mm Particle Content	%	≤1	0.2
Water Absorption	%	≤2.0	0.9
Durability	%	≤12	Pass
Clay Content	%	≤1	0.5

**Table 5 polymers-16-02766-t005:** Technical indexes of fine aggregate.

Test Parameters	Units	Technical Standards	Test Results
Apparent Relative Density	—	≥2.50	2.708
Sand Equivalent	%	≥60	64
Durability (Particles >0.3 mm)	%	≥12	Pass
Clay Content (Content <0.075 mm)	%	≤3	1.2

**Table 6 polymers-16-02766-t006:** Technical indexes of mineral powder.

Test Parameters	Units	Technical Standards	Test Results
Apparent Density	t/m^3^	≥2.50	2.6
Sieve Analysis (Particles <0.075 mm)	%	75~100	90.2
Moisture Content	%	≤1	0.3
Hydrophilicity Coefficient	—	≤1	0.68
Heat Stability (200 °C)	—	Measured Data	No Change in Color

**Table 7 polymers-16-02766-t007:** Segregation test results of TPU-modified asphalt.

Asphalt Type	S1 (Softening Point of the Top, °C)	S2 (Softening Point of the Bottom, °C)	ΔS (Softening Point Difference, °C)
Base asphalt	46.8	47.0	0.2
5%TPU	51.6	52.2	0.6
10%TPU	54.1	55.0	0.9
15%TPU	56.9	57.8	0.9
20%TPU	57.0	58.0	1.0

**Table 8 polymers-16-02766-t008:** Segregation test results of SBR-modified asphalt.

Asphalt Type	S1 (Softening Point of the Top, °C)	S2 (Softening Point of the Bottom, °C)	ΔS (Softening Point Difference, °C)
2.0%SBR	47.7	48.4	0.7
2.5%SBR	47.8	48.7	0.9
3.0%SBR	48.1	49.3	1.2
3.5%SBR	49.4	50.8	1.4
4.0%SBR	49.7	51.5	1.8

## Data Availability

The original contributions presented in the study are included in the article, further inquiries can be directed to the corresponding author.

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
