# Peer review of "Study on the Microscopic Mechanism and Performance of TPU/SBR Composite-Modified Asphalt"

_polymers, 2024, doi:10.3390/polym16192766_

Round 1
Reviewer 1 Report
Comments and Suggestions for Authors
This study focused on the properties of composite-modified asphalt binder with TPU and SBR. Various properties related to material characteristics and pavement performance were studied and presented. The following are comments for this paper;
1- The authors have to significantly improve the quality of the English language. There are many sentence-forming errors, spelling mistakes, and errors in units. This should improve significantly.
2- Please improve your introduction with proper description on novelty of this study. Already many similar studies are there. What is new in this research? Please highlight clearly.
3- Please improve the quality of Fig. 6 and Fig. 7. the discussion on the results should improve by providing appropriate citations to substantiate the findings of this study. This is missing in the manuscript.
4- Why there is not much difference in softening point for SBR mixes. Please explain
5- Please correct the units on line 340 and 341
6- Why there is a difference in the segregation test results of TPU modified when compared to SBR modified? Please provide a better explanation with relevant papers cited for reasoning.
7- Improve the quality of Fig.8 . It is difficult to interpret. there is an observed difference in elastic recovery for both the modified asphalt. Please provide a better explanation for these findings.
8- Please improve conclusion. Also provide challenges and future scope for this research.
Comments on the Quality of English LanguageEnglish Language should be significantly improved.
Author Response
1- The authors have to significantly improve the quality of the English language. There are many sentence-forming errors, spelling mistakes, and errors in units. This should improve significantly.
Response: Thank you for the detailed feedback from the reviewer. We highly value your feedback and have taken measures to comprehensively improve the language and format of the manuscript. In order to improve the quality of the English language, several students in our research group have carefully proofread and revised the manuscript multiple times. They have reviewed each language error in the manuscript and made necessary corrections, including correcting sentence structure errors, spelling errors, and unit expression errors. All language errors and formatting issues have been corrected, and we really hope to meet the publishing requirements.
2- Please improve your introduction with proper description on novelty of this study. Already many similar studies are there. What is new in this research? Please highlight clearly.
Response: Thank you for your comment. According to your suggestion, we have
emphasized the novelty of the work at the end of the introduction. The specific contents of the end of the introduction are as follows:
Current research indicates that polyurethane, as a chemically modified material, contains functional groups capable of reacting with and bonding to asphalt. This interaction results in a stable dispersion within the asphalt matrix and enhances the cross-linking effect among asphalt molecules. Consequently, polyurethane significantly improves the high-temperature performance and storage stability of asphalt and its mixtures, demonstrating promising potential for engineering applications. However, polyurethane may adversely affect the low-temperature performance of asphalt, leading to increased brittleness and a higher propensity for cracking during winter conditions, which compromises the structural integrity of asphalt pavements. Therefore, it is essential to identify a complementary modifier that can be blended with polyurethane to address these low-temperature performance issues effectively while also providing economic benefits for composite modification.Based on an analysis of the current research both domestically and internationally, Styrene-Butadiene-Rubber (SBR) is known to significantly enhance the low-temperature performance and fatigue resistance of asphalt and its mixtures. However, its limitations include a lack of notable improvement in high-temperature performance and inadequate storage stability, which constrain its application in regions with high summer temperatures. Consequently, there has been considerable research into the composite modification of asphalt by combining SBR with other modifiers, such as Montmorillonite (MMT). While these combinations can address the deficiencies of SBR-modified asphalt in high-temperature conditions, they have not effectively resolved the issue of storage stability. Given the performance advantages of Thermoplastic Polyurethane (TPU) and Styrene-Butadiene-Rubber (SBR), leveraging the physical and chemical blending effects of their combination could significantly enhance the performance and service life of asphalt. However, there is currently a lack of systematic research on the use of TPU and SBR in composite modification of asphalt. This gap includes the absence of a comprehensive performance evaluation system and a material design theory, which impedes the practical application and further development of such composite modified asphalt.This study investigates the composite modification of matrix asphalt using Thermoplastic Polyurethane (TPU) and Styrene-Butadiene-Rubber (SBR). Specifically, TPU and SBR are employed as individual modifiers to produce TPU-modified asphalt and SBR-modified asphalt, respectively. The research involves basic performance testing of these asphalts to analyze the impact of each modifier on asphalt properties. The optimal dosages of TPU and SBR are determined through these tests, and TPU/SBR composite modified asphalt is prepared accordingly. The study evaluates storage stability, elastic recovery, microstructure, and modification mechanisms of the modified asphalt using standard experimental methods. Additionally, the influence of the TPU/SBR combination on the properties of the modified asphalt is explored. We hope to develop a composite modified asphalt with more comprehensive performance, providing ideas for the research and development of high-performance and long-life materials for road surfaces.(line109-121;131-139;150-152)
3- Please improve the quality of Fig. 6 and Fig. 7. the discussion on the results should improve by providing appropriate citations to substantiate the findings of this study. This is missing in the manuscript.
Response: We have redrawn Fig.6 and Fig.7 to ensure their accuracy and clarity, and have also appropriately cited cutting-edge papers in the article to validate the viewpoints, making the discussion of the results more rigorous and complete.Our changes have been highlighted in red in the text. (line355-356;358-359;365-367;379-386)
Fig.6. Physical properties of TPU modified asphalt
Fig.7. Physical properties of SBR modified asphalt
4- Why there is not much difference in softening point for SBR mixes. Please explain
Response:Thank you for your comment. Based on your suggestion, we have explained why there is not much difference in softening point for SBR mixes.The specific contents are as follows:
The changes of penetration and softening point were not obvious, SBR has limited improvement on the high-temperature performance of matrix asphalt, while pure SBR modified asphalt exhibits poor adhesion to aggregates and poor compatibility with asphalt, making it unstable at high temperatures.
(line379-383;546-555)
5- Please correct the units on line 340 and 341
Response:Thank you for your comment. Regarding the units mentioned on lines 340 and 341, I have carefully reviewed and corrected them as suggested. Specifically, "I corrected the unit from 'm' to ' mm' . I have also conducted a thorough review of the manuscript to ensure that all units are correctly applied.
(line376-377)
- Why there is a difference in the segregation test results of TPU modified when compared to SBR modified? Please provide a better explanation with relevant papers cited for reasoning.
Response:Thank you for your comment. Based on your suggestion, we have explained why the segregation test results of modified TPU differ from those of modified SBR and cited relevant papers for inference.The specific contents are as follows:
The reason why the softening point difference increases with the increase of SBR content may be that SBR, as a synthetic rubber, has certain differences in solubility parameters and density components compared to asphalt. Even after high-temperature and high-speed shear mixing and dispersion, SBR is difficult to be fully compatible with the base asphalt, and may even lead to partial aggregation of heavy components in the asphalt. At the same time, the polarity characteristics of SBR modified asphalt will also change accordingly, resulting in an increase in the softening point difference value with the increase of SBR content, which also means that the storage stability of SBR modified asphalt is poor. However, TPU modified asphalt did not undergo segregation, and the compatibility between TPU modifier and asphalt was good, indicating that TPU modified asphalt has excellent storage stability. Related studies have shown that the storage stability of different materials mainly depends on their polarity differences. The greater the polarity difference, the poorer the storage stability, while the smaller the polarity difference, the better the storage stability. TPU modifier does not condense during high-temperature storage, thereby minimizing the migration and separation of the modifier. The main reason is the reaction between the - NCO group in TPU and the active hydrogen atoms (mainly - OH) in asphaltene micelles, which reduces the polarity difference between material components and effectively improves the compatibility between TPU and matrix asphalt.
(line546-569)
7- Improve the quality of Fig.8 . It is difficult to interpret. there is an observed difference in elastic recovery for both the modified asphalt. Please provide a better explanation for these findings.
Response:Thank you for your comment.Based on your suggestion, Fig.8 has been redrawn to enhance its accuracy and clarity, as follow. As observed in Fig.9 and Fig.10, TPU-modified asphalt exhibits a higher elastic recovery rate, with a maximum difference of 16.9. In contrast, the elastic recovery rate of SBR-modified asphalt remains relatively unchanged, showing a maximum difference of 4.7. This indicates that TPU has a more pronounced effect on improving the elastic recovery performance of asphalt compared to SBR. In addition, we analyzed the variation in the asphalt phase angle (δ) with temperature. This phase angle reflects the stress-strain behavior of asphalt under external loads and provides insight into the relative proportions of viscosity and elasticity within the asphalt.These contents should better explain the differences in elastic recovery between TPU modified asphalt and SBR modified asphalt.The specific extensions are as follows:
Fig.8. Elastic recovery rate of TPU modified asphalt
Fig.13. Curve of asphalt phase angle with temperature
The phase angle δ reflects the changes in stress and strain of asphalt under external loads, and can also reflect the proportional relationship between the two components of asphalt viscosity and elasticity. The larger the phase angle δ value, the more viscous components there are in asphalt than elastic components, which to some extent improves the flow and deformation ability of asphalt, while the deformation recovery ability weakens, manifested as asphalt pavement being more prone to rutting diseases[50]. The temperature scanning test results of 7 different types of asphalt are shown in Fig.13. It can be seen that the phase angle δ of asphalt is positively correlated with temperature, that is, as the temperature increases, the phase angle δ also increases, and the viscosity component also increases. Asphalt is more prone to softening under high temperature conditions, leading to permanent deformation. Under the same temperature conditions, the phase angle δ of the matrix asphalt is the largest, showing a size pattern of 70 #>3.5% SBR>3% SBR+10% TPU>3.5% SBR+10% TPU>15% TPU>3% SBR+15% TPU>3.5% SBR+15% TPU. This indicates that the addition of modifiers causes significant changes in the viscoelastic composition of asphalt, with a decrease in viscous components and an increase in elastic components, thereby improving the elastic recovery ability of asphalt and effectively improving high-temperature rutting resistance. In addition, at the same temperature, the phase angle δ of the four types of composite modified asphalt is smaller than that of SBR modified asphalt. Among the composite modified asphalt, the TPU content has the most significant effect on its phase angle δ, indicating that the addition of TPU to SBR modified asphalt effectively reduces the proportion of viscous components and increases the proportion of elastic components, thereby enhancing the high-temperature deformation resistance of asphalt. When TPU and SBR are used for composite modification of matrix asphalt, the interaction between the two modifiers significantly enhances the deformation recovery ability and high-temperature rutting resistance of the composite modified asphalt.
(line585-611)
8- Please improve conclusion. Also provide challenges and future scope for this research.
Response:Thank you for your comment. In response to your suggestions, we have revised the conclusion to more effectively summarize the key findings and implications. Additionally,I have made some suggestions for future research. The updated conclusions section reads as follows:
This study investigates the modification of 70# matrix asphalt using Thermoplastic Polyurethane (TPU) and Styrene-Butadiene-Rubber (SBR) as modifiers. Initially, routine performance tests were conducted to assess the physical properties, storage stability, and elastic recovery of TPU-modified and SBR-modified asphalts at various dosages. Based on these technical performance indicators, the optimal dosages for both modifiers were determined. Subsequently, TPU/SBR composite modified asphalt was prepared, and its performance changes were analyzed. Using atomic force microscopy, fluorescence microscopy, and Fourier transform infrared spectroscopy, the microstructure, phase structure, and modification mechanisms of the asphalt before and after modification were examined, with an in-depth analysis of the underlying reasons for performance changes. The main conclusions are as follows:
(1)TPU increases the viscosity of asphalt, with a more pronounced effect at higher dosages. It exhibits good compatibility with asphalt, leading to enhanced high-temperature performance and storage stability. However, ductility slightly decreases, and low-temperature performance is somewhat reduced. Based on conventional performance, a recommended TPU dosage is 15%.
(2) SBR significantly enhances the flexibility of asphalt, thereby improving its low-temperature performance. However, as the dosage increases, compatibility with asphalt decreases, resulting in poor storage stability. Based on conventional performance, a recommended SBR dosage is 3.5%.
(3) The addition of TPU and SBR improves the conventional performance of asphalt. Compared to TPU-modified asphalt, the ductility of the composite modified asphalt is significantly increased, indicating that SBR enhances the low-temperature ductility of TPU-modified asphalt. TPU improves the viscosity and compatibility of SBR-modified asphalt, thus enhancing its high-temperature performance and storage stability.
(4) TPU facilitates the even dispersion of the heavy components in SBR-modified asphalt into lighter components, improving its structural composition. It forms a more stable cross-linked network structure, effectively constraining molecular movement under high-temperature conditions and enhancing high-temperature deformation resistance.
(5) Matrix asphalt initially has a single-phase structure. The addition of TPU and SBR transforms it into a dispersed phase structure. The good compatibility among the components results in TPU/SBR composite modified asphalt exhibiting excellent storage stability.
(6) SBR undergoes a physical reaction with asphalt, significantly enhancing the low-temperature performance of asphalt through physical blending. TPU modified asphalt exhibits distinct new characteristic absorption peaks at 1730, 1191, and 1177, indicating a chemical reaction between the - NCO group of TPU and certain polar groups of asphalt (such as - OH). Therefore, the composite modified asphalt system undergoes both physical and chemical modifications, resulting in excellent comprehensive performance.
5 The limitations and future scope of the study
This study systematically investigates the conventional properties and microscopic characteristics of TPU/SBR composite modified asphalt. However, due to limitations in my academic capacity, a comprehensive analysis of the properties of TPU/SBR composite modified asphalt remains incomplete. It is recommended that future research address the following areas:
(1) The impact of various preparation processes on asphalt performance, including but not limited to shear temperature, rate, and duration. Variations in these factors may lead to significant changes in asphalt performance, warranting more detailed investigation.
(2) An exploration of the effects of thermal oxidation and ultraviolet aging on the performance of TPU/SBR composite modified asphalt. Additionally, research should examine the performance changes of TPU/SBR composite modified asphalt and its mixtures before and after aging.
(line835-891)
Reviewer 2 Report
Comments and Suggestions for Authors
Although the topic of this article is interesting and significant to the scientific community, but I would like to take minor review from the authors before acceptance. The comments on this article that is to be answered are:
The abstract should be redone. There are currently no specific results obtained. There is only a mention of the optimal amount of TPU/SBR. It is necessary to add the values of the performance characteristics of the asphalt at these optimal values. The abstract is a mini version of the manuscript that proceeds. So, include the introduction, methodology, results and concluding remarks in a precise but effective manner.
2. Table 2 needs to be brought to a similar form to tables 1, 3-6. These tables indicate Technical standard and Measured results. But in table 2 it is not clear what it is. In addition, if tables 1, 3-6 present the results measured by the authors, it is necessary to fully describe the equipment and methods by which they carried out the measurements indicated in the tables.
The authors indicate the following wavelength range for FTIR spectroscopy: 40~40cm-1. This is clearly not true.
3. Section 2 should indicate the devices used to conduct the studies. The brands of the IR-Fourier spectrometer and fluorescence microscope are not listed in Section 2. The device is listed when describing the fluorescence analysis, but the IR-Fourier spectrometer is not listed at all throughout the article.
4. There are no measurement errors in figures 6-9. Although the values​are close to each other. Perhaps the results obtained are within the confidence interval, this all needs to be rechecked.
5. Future work can also be included in the conclusion section.
Author Response
1.The abstract should be redone. There are currently no specific results obtained. There is only a mention of the optimal amount of TPU/SBR. It is necessary to add the values of the performance characteristics of the asphalt at these optimal values. The abstract is a mini version of the manuscript that proceeds. So, include the introduction, methodology, results and concluding remarks in a precise but effective manner.
Response:Thank you for your comment. We have reorganized the abstract section of this article and provided a comprehensive summary of the specific results. The rewritten summary is as follows:
To enhance the service life of traditional asphalt pavement and mitigate issues such as high-temperature rutting and low-temperature cracking, this study investigates the composite modification of matrix asphalt using Thermoplastic Polyurethane (TPU) and Styrene-Butadiene-Rubber (SBR). Initially, the study examines the conventional properties of the composite modified asphalt from a macro perspective, analyzing the performance variations of asphalt before and after TPU and SBR modification. Subsequently, a microscopic analysis is conducted to explore the microstructure, phase structure, and modification mechanisms of the composite modified asphalt, with a focus on understanding the underlying reasons for performance changes.The influence of TPU and SBR on asphalt performance is evaluated comprehensively. It is found that TPU-modified asphalt demonstrates superior high-temperature performance, storage stability, and elastic recovery. Conversely, SBR-modified asphalt excels in ductility at low temperatures, though its storage stability decreases with increasing dosage. Based on a thorough analysis of the conventional properties of the two types of modified asphalt, the optimal dosages of TPU and SBR are determined to be 15% and 3.5%, respectively. In the composite modified asphalt, TPU facilitates the even distribution of chemical components, creating a more stable cross-linked network structure. The compatibility of TPU, SBR, and asphalt contributes to the good storage stability of the composite modified asphalt. While SBR effects physical modification, TPU induces chemical modification of asphalt. Consequently, the composite modification system benefits from both physical and chemical enhancements, resulting in excellent overall performance.
(line8-29)
- Table 2 needs to be brought to a similar form to tables 1, 3-6. These tables indicate Technical standard and Measured results. But in table 2 it is not clear what it is. In addition, if tables 1, 3-6 present the results measured by the authors, it is necessary to fully describe the equipment and methods by which they carried out the measurements indicated in the tables.The authors indicate the following wavelength range for FTIR spectroscopy: 40~40cm-1. This is clearly not true.
Response:Thank you for your comment. We have listed the technical standards and measurement results in Table 2. The equipment and methods for TPU measurement were referenced from other literature. Although specific technical standards, equipment, and methods were not provided, we cited them in the article, as follows:
Polyurethane is a high molecular weight polymer with interlocking soft and hard segments [23, 24]. According to different processing methods and molecular structures, polyurethane can be divided into two types: thermoplastic polyurethane and thermosetting polyurethane [25, 26]. The specific type of polyurethane selected in this article is thermoplastic polyurethane, which is white in color and appears as a powdery solid.
Table 2 Properties of TPU
|
Test Parameters |
Units |
Test Results |
|
Appearance |
— |
White Particles |
|
Particle Size |
Mesh |
32 |
|
Density |
g/cm3 |
1.11 |
|
Tensile Strength |
MPa |
55.3 |
|
Elongation at Break |
% |
479.1 |
(line163-176)
- Section 2 should indicate the devices used to conduct the studies. The brands of the IR-Fourier spectrometer and fluorescence microscope are not listed in Section 2. The device is listed when describing the fluorescence analysis, but the IR-Fourier spectrometer is not listed at all throughout the article.
Response:Thank you for your comment. According to your recommendations,We provided a comprehensive description of the equipment used in the research, including the brands and experimental parameters for Atomic force microscopy (AFM) test,Fourier transform infrared spectroscopy (FTIR) test and Fluorescence microscope (FM) test, as follows:
Atomic force microscope (AFM) test can analyze the micro surface morphology, viscosity and molecular interaction of polymer modified asphalt[27-37]. In this study, FastScan atomic force microscope is used(from BRUKER Corporation in the United States). The imaging mode is tap type, and the scanning area is set as 20 μm × 20 μm. Two dimensional and three dimensional morphologies of different asphalt materials were obtained by scanning. Evaluate the dissolution and dispersion of modifiers in asphalt by analyzing the area, quantity, and height of micro surface honeycomb structures of different asphalt materials. And analyze the compatibility between the modifier and asphalt to determine the stability of the newly formed modification system.
Fourier transform infrared spectroscopy (FTIR) test can analyze the chemical composition and functional groups of polymer modified asphalt[27-36]. The instrument used is a VERTEX 70 Fourier Transform Infrared Spectrometer (sourced from Bruker GmbH in Germany). During the experiment, a scanning resolution of 4and a wavelength range of 4000~400 were used, with 32 scans for each type of asphalt. Place the prepared asphalt in the corresponding position of the infrared spectrometer, use a computer to collect data and obtain corresponding infrared spectra. Based on the characteristic peaks of different asphalt samples, the molecular structure and functional groups of the matrix asphalt and modified asphalt can be analyzed.
Fluorescence microscopy is based on the human eye can not see a certain wavelength of high-energy ultraviolet light to irradiate the material. The instrument is selected from IMAGER.Z2 fluorescence microscope (sourced from Carl Zeiss Optics GmbH in Germany). At present, the FM test sample preparation method adopts the hot drop cover glass method for sample preparation. The magnification of the sample observation is 100 times.
(line296;309-317;332-336)
- There are no measurement errors in figures 6-9. Although the valuesare close to each other. Perhaps the results obtained are within the confidence interval, this all needs to be rechecked.
Response:Thank you for your comment. We have carefully examined Fig.6-Fig.9 to ensure that there are no measurement errors present.
- Future work can also be included in the conclusion section.
Response:Thank you for your comment. In response to your suggestions, we have further discussed the conclusion section and made some suggestions for future research. The updated conclusions section reads as follows:
This study investigates the modification of 70# matrix asphalt using Thermoplastic Polyurethane (TPU) and Styrene-Butadiene-Rubber (SBR) as modifiers. Initially, routine performance tests were conducted to assess the physical properties, storage stability, and elastic recovery of TPU-modified and SBR-modified asphalts at various dosages. Based on these technical performance indicators, the optimal dosages for both modifiers were determined. Subsequently, TPU/SBR composite modified asphalt was prepared, and its performance changes were analyzed. Using atomic force microscopy, fluorescence microscopy, and Fourier transform infrared spectroscopy, the microstructure, phase structure, and modification mechanisms of the asphalt before and after modification were examined, with an in-depth analysis of the underlying reasons for performance changes. The main conclusions are as follows:
(1)TPU increases the viscosity of asphalt, with a more pronounced effect at higher dosages. It exhibits good compatibility with asphalt, leading to enhanced high-temperature performance and storage stability. However, ductility slightly decreases, and low-temperature performance is somewhat reduced. Based on conventional performance, a recommended TPU dosage is 15%.
(2) SBR significantly enhances the flexibility of asphalt, thereby improving its low-temperature performance. However, as the dosage increases, compatibility with asphalt decreases, resulting in poor storage stability. Based on conventional performance, a recommended SBR dosage is 3.5%.
(3) The addition of TPU and SBR improves the conventional performance of asphalt. Compared to TPU-modified asphalt, the ductility of the composite modified asphalt is significantly increased, indicating that SBR enhances the low-temperature ductility of TPU-modified asphalt. TPU improves the viscosity and compatibility of SBR-modified asphalt, thus enhancing its high-temperature performance and storage stability.
(4) TPU facilitates the even dispersion of the heavy components in SBR-modified asphalt into lighter components, improving its structural composition. It forms a more stable cross-linked network structure, effectively constraining molecular movement under high-temperature conditions and enhancing high-temperature deformation resistance.
(5) Matrix asphalt initially has a single-phase structure. The addition of TPU and SBR transforms it into a dispersed phase structure. The good compatibility among the components results in TPU/SBR composite modified asphalt exhibiting excellent storage stability.
(6) SBR undergoes a physical reaction with asphalt, significantly enhancing the low-temperature performance of asphalt through physical blending. TPU modified asphalt exhibits distinct new characteristic absorption peaks at 1730, 1191, and 1177, indicating a chemical reaction between the - NCO group of TPU and certain polar groups of asphalt (such as - OH). Therefore, the composite modified asphalt system undergoes both physical and chemical modifications, resulting in excellent comprehensive performance.
5 The limitations and future scope of the study
This study systematically investigates the conventional properties and microscopic characteristics of TPU/SBR composite modified asphalt. However, due to limitations in my academic capacity, a comprehensive analysis of the properties of TPU/SBR composite modified asphalt remains incomplete. It is recommended that future research address the following areas:
(1) The impact of various preparation processes on asphalt performance, including but not limited to shear temperature, rate, and duration. Variations in these factors may lead to significant changes in asphalt performance, warranting more detailed investigation.
(2) An exploration of the effects of thermal oxidation and ultraviolet aging on the performance of TPU/SBR composite modified asphalt. Additionally, research should examine the performance changes of TPU/SBR composite modified asphalt and its mixtures before and after aging.
(line835-891)

Round 2
Reviewer 1 Report
Comments and Suggestions for Authors
The authors have addressed all my comments. Minor corrections to the paper format and English language are necessary before publication.